# Not Only Graphene Two-Dimensional Nanomaterials: Recent Trends in Electrochemical (Bio)sensing Area for Biomedical and Healthcare Applications

**DOI:** 10.3390/molecules29010172

**Published:** 2023-12-27

**Authors:** Paola Di Matteo, Rita Petrucci, Antonella Curulli

**Affiliations:** 1Dipartimento Scienze di Base e Applicate per l’Ingegneria, Sapienza University of Rome, 00161 Rome, Italy; p.dimatteo@uniroma1.it (P.D.M.); rita.petrucci@uniroma1.it (R.P.); 2Consiglio Nazionale delle Ricerche (CNR), Istituto per lo Studio dei Materiali Nanostrutturati (ISMN), 00161 Rome, Italy

**Keywords:** electrochemical (bio)sensors, 2D nanomaterials, TMDs, MOFs, COFs, MXenes, biomedical analysis, healthcare

## Abstract

Two-dimensional (2D) nanomaterials (e.g., graphene) have attracted growing attention in the (bio)sensing area and, in particular, for biomedical applications because of their unique mechanical and physicochemical properties, such as their high thermal and electrical conductivity, biocompatibility, and large surface area. Graphene (G) and its derivatives represent the most common 2D nanomaterials applied to electrochemical (bio)sensors for healthcare applications. This review will pay particular attention to other 2D nanomaterials, such as transition metal dichalcogenides (TMDs), metal–organic frameworks (MOFs), covalent organic frameworks (COFs), and MXenes, applied to the electrochemical biomedical (bio)sensing area, considering the literature of the last five years (2018–2022). An overview of 2D nanostructures focusing on the synthetic approach, the integration with electrodic materials, including other nanomaterials, and with different biorecognition elements such as antibodies, nucleic acids, enzymes, and aptamers, will be provided. Next, significant examples of applications in the clinical field will be reported and discussed together with the role of nanomaterials, the type of (bio)sensor, and the adopted electrochemical technique. Finally, challenges related to future developments of these nanomaterials to design portable sensing systems will be shortly discussed.

## 1. Introduction

The development of (bio)sensors for the quick and cost-effective determination of low-level analytes within a wide linearity range can be considered a key goal to be achieved in the field of biomedical analysis and healthcare [1,2,3], among others.

The main requirement to compete with conventional analytical methods is the realization of an accurate and robust analysis system through the development of sensing platforms capable of stabilizing the biorecognition element, if present, and effectively interweaving with the signal transduction system. Consequently, the selection of appropriate sensing materials is fundamental for achieving the required performance.

Among various (bio)sensors, the electrochemical ones can represent smart detection tools for biomedical analysis and healthcare as part of accurate, sensitive, specific, and rapid analysis systems [4,5].

Recently, the interest in two-dimensional (2D) nanomaterials has grown considerably, and one-atom-thick graphene (G) is definitely the best known and most studied [6].

Due to its peculiar physicochemical properties, the so-called ‘graphene rush’ has triggered the study of atomically thin sheets of other layered materials, such as transition metal dichalcogenides (TMDs), metal–organic frameworks (MOFs), covalent organic frameworks (COFs), and MXenes, among others.

The chemical and structural heterogeneity of these new 2D nanomaterials, whose properties are actually controlled and tuned by their dimensions, provides huge possibilities for basic and applied research. In particular, the high surface area-to-volume ratio together with their optical/electrical properties and biocompatibility make such materials extremely attractive for their application in sensing and biosensing areas, including electrochemical sensors and biosensors such as immunosensors, genosensors, enzyme-based biosensors, and aptasensors.

This review focuses on the application of various 2D nanomaterials beyond graphene in electrochemical biosensing and sensing systems for the analysis of significant analytes of clinical and biomedical interest, such as glucose, neurotransmitters, hormones, viruses, cancer biomarkers, and drugs, among others.

Several reviews reported in the literature are focused on the application of 2D nanomaterials in the (bio)sensing area [1,2,6,7,8,9,10,11,12,13], but no special attention has been paid to electrochemical (bio)sensors for biomedical and healthcare applications.

This review aims to provide an up-to-date overview of which 2D nanomaterials beyond G are applied to electrochemical (bio)sensors for biomedical and healthcare applications, considering the literature of the last five years (2018–2022), evidencing strengths, limits, and future perspectives. For this purpose, it is organized in two parts.

In the first part, the synthetic approach and properties of 2D nanomaterials used in the herein reported (bio)sensor examples are briefly described; the second part introduces the most significant examples regarding the determination of analytes of clinical and biomedical interest.

Special attention is paid to the role of the involved nanomaterial, type of receptor, recognition mechanism, and selectivity, beyond the possibility of using them on real samples, and to the comparison with official validation methods, if provided.

Brief comments and/or observations are reported at the end of each subsection, but a more detailed and in-depth discussion can be found in Section 4.

## 2. Two-Dimensional Nanomaterials

It is well-known that dimensions of 2D nanomaterials are not within the nanoscale and that they are considered single- or few-layer nanocrystalline materials with a planar morphology. Electronic conduction is restricted by the nanostructure’s thickness and it is assumed delocalized on the plan of the sheet [2,4]. They make up a big family involving different elements, from transition metals to carbon, nitrogen, and/or sulfur [12]. Among them, G is the most popular for designing and developing electrochemical (bio)sensors, drawing attention to other 2D nanomaterials.

More recently, graphene-like 2D-layered nanomaterials such as MXenes, TMDs, COFs, and MOFs have evidenced strong mechanical strength, high surface areas, fast electron transfer kinetics, significant biocompatibility, and easy functionalization; for these reasons they have begun to be used to assemble and develop electrochemical sensors.

### 2.1. MXenes

MXenes display peculiar properties, including metallic electrical conductivity [14,15] and hydrophilicity, which are not always available in other 2D family members.

MXenes are 2D transitional metal carbides/nitrides/carbonitrides, obtained by the selective etching of the so-called MAX phases (Figure 1). MAX phases are conductive 2D layers of transition metal carbides/nitrides interconnected by the ‘element A’ (Group 13–16) through strong ionic, metallic, and covalent bonds [14,15]. In chemical exfoliation, method hydrofluoric acid (HF) or a mixture of lithium fluoride and hydrochloric acid (HCl) are used, while the electrochemical selective exfoliation procedure is assumed as a fluoride-free etching procedure [14].

M_n+1_X_n_T_x_ is the general formula, M being an early transition metal (Mo, W, Ti, V, Sc, Y, Zr, Hf, Nb, Ta, or Cr) and X carbon or nitrogen with n between 1 and 3. T indicates surface functional groups, such as -O, -F, -OH, and rarely -Cl and x in T_x_ their number. MXenes are classified as layered sheet-like nanomaterials and their thicknesses, starting from 1 nm, can be modulated to vary in value in MXenes’ general formula to provide different chemical compositions, using both computational and experimental methods.

MXenes have been intensively investigated for designing and assembling electrochemical (bio)sensors because of their high electrochemical activity and conductivity, and for their large surface area [16,17,18,19,20]. Notably, MXenes can be easily functionalized with other materials and their performance in the electrochemical sensing area relies on the type and number of functional groups [16,20]. MXenes nanohybrids can be synthesized including, for instance, metal nanoparticles and other nanomaterials, with a synergistic effect enhancing the sensing properties.

Notably, novel 2D monoelemental materials called Xenes, (e.g., borophene, silicene, germanene, stanene, phosphorene, arsenene, antimonene, bismuthene, and tellurene) have recently attracted considerable attention as promising 2D nanomaterials for biosensors, bioimaging, therapeutic delivery, theranostics, and other new bio-applications, due to their interesting physical, chemical, electronic, and optical properties [21].

When the lateral dimension of 2D nanomaterial is lower than 10 nm, 2D quantum dots (2D-QDs) can be produced owing to the strong quantum confinement effect [22,23]. Thus, 2D-QDs are generally classified as quantum dots derived from 2D materials. Although 2D-QDs might be considered zero-dimensional (0D) nanomaterials, they can actually be assumed as miniaturized/scaled-down structures of 2D-layered materials retaining their two-dimensional lattices. Compared with conventional forms, they present better solubility and dispersibility, beyond a larger surface area-to-volume ratio, so they can be easily doped and functionalized. In addition, they generally present electrochemical activity and high electrical conductivity. All these properties make 2D-QDs suitable for applications in various fields, including electrochemical sensing and biosensing areas, and appealing for the development of electrochemical sensors.

Two-dimensional quantum dots are generally synthesised by top-down approaches, involving the cleavage of bulk precursors such as the 2D nanomaterial, and by bottom-up approaches, including the aggregation and growth of small organic or inorganic molecules [22,23].

Regarding 2D-QDs derived from MXenes, the ultra-thin size involves a large specific surface area and a high number of functional groups, thereby allowing the binding of a high number of biomolecules and/or interacting more effectively with different analytes.

### 2.2. TMDs

TMDs represent a very interesting and unique 2D nanomaterial’s family. Their general formula is MX_2_, where M is a transition metal such as Ti, Zr, V, Nb, Mo, W, Hf, and Ta and X is a chalcogen such as Se, S, or Te [24,25]. TMDs are characterized by an M-X-M sandwich structure, in which the chalcogen X and the two metal atoms M in the same layer are covalently linked. Different layers are held together only by weak van der Waals interactions (Figure 2).

According to their electronic band structure [26], TMDs can be defined as insulators (such as HfS_2_), semiconductors (such as MoS_2_ and WS_2_), semi metals (such as WTe_2_ and TiSe_2_), and metals (such as NbS_2_ and VSe_2_).

Due to their physical, chemical, optical, and electronic features, such as high bandgap, large surface area-to-volume ratio, large number of active sites available for further functionalization, good electrical conductivity, and fast electron transfer, TMDs can be considered promising nanomaterials to be applied in the electrochemical sensing area [26,27].

Different synthetic approaches have been reported, such as mechanical cleavage, liquid and chemical exfoliation, chemical vapour deposition (CVD), and hydrothermal/solvothermal processes. They can be classified as top-down or bottom-up methods [26].

Starting from the top-down approaches, the mechanical cleavage is the most common method, involving mechanical exfoliation by using scotch tape or adhering polymeric thin films to obtain TMDs single-layer or multi-layer structures from bulky materials [28,29]. It is an easy and inexpensive technique, but it is time demanding, and quality and thickness are not properly managed.

In the chemical exfoliation, intercalators (organometallic compounds) are introduced inside the bulk material via ultrasound. This method develops different ultrathin TMDs without involving toxic solvents. However, it is time demanding and requires a high reaction temperature and high environmental awareness.

The liquid intercalation and exfoliation involve a stabilizer/dispersing agent and solvent. The liquid exfoliation requires three typical steps: bulk starting the material dispersion in solvent, exfoliation, and purification for isolating TMD-exfoliated layers from the unexfoliated ones. It is an easy and cost-effective method, and single-layer or multi-layer nanostructures can be produced on a large scale. Conversely, using toxic solvents can be considered a real disadvantage.

Moving to the bottom-up techniques, the hydrothermal and solvothermal approaches are conventional methods for the synthesis of colloidal TMDs.

The solvothermal method requires high boiling organic solvents and efficient nucleation and growth procedures. The presence of organic ligands is mandatory for controlling TMDs’ morphology and size, and for guaranteeing dispersibility. The difference with the hydrothermal method is the precursor is not in an aqueous solution. Hydrothermal and solvothermal approaches are simple and easy to apply but not suitable for large-scale production.

Finally, the bottom-up CVD method under high temperature and pressure conditions produces TMDs’ high-quality layers on different substrates. Chalcogenide atoms and transition metals are provided by the corresponding precursors. TMD films with peculiar electronic properties and optimized thicknesses were synthesized by means of the bottom-up CVD method. However, high-temperature and high-vacuum conditions are required for the synthetic protocol.

Two-dimensional quantum dots derived from TMDs [22,23] show large specific surface areas and a high number of functional groups, so they can be used for electrochemical sensors and biosensors, as already reported in Section 2.1 for 2D-QDs derived from MXenes.

### 2.3. MOFs

Metal–organic frameworks (MOFs) are assumed as organic–inorganic hybrid crystalline porous coordination polymers. They are assembled with inorganic and organic units, acting as ligands through coordination binding, to realize a network together with a flexible and tunable structure [30,31]. MOFs can represent very promising electrode materials for different application fields, since the selection of MOFs’ “ingredients” will determine the geometry, form, and size of the pores, as well as the surface area and functionality of the structure. However, the most common MOFs have low electrical conductivity and relatively poor stability under electrochemical experimental conditions. To solve this critical issue, MOF-hybrid electrochemical sensing materials have been developed introducing metal nanoparticles (MNPs) or metal oxide nanoparticles (MONPs) in the corresponding reticulated structure.

Different strategies have been applied for the synthesis of MOFs (Figure 3). In addition, post-synthetic methods for surface functionalization and the modification of MOFs have been implemented during the last years [32,33]. The most common methods are reported and discussed below.

The solvothermal (or hydrothermal) method is a conventional procedure to synthesize MOFs crystalline structures, performed at the solvent boiling temperature, as previously reported for TMDs in Section 2.2. A direct coordination between organic ligands and metal clusters is obtained and the crystals’ nucleation/growth rate is controlled by the applied temperature, assuring coordination between the inorganic and organic units, irrespective of the generated pressure. However, this method is time and energy demanding.

In the microwave-assisted method, the microwave irradiation supports MOF crystals’ nucleation: growth rate, different morphologies, and size, together with faster crystallization, can be achieved by controlling the key parameters such as solvent, energy power, reactants concentration, and reaction time.

With regard to the electrochemical strategy, the anode and cathode are immersed in the electrolytic solution in the presence of the organic linker and the inorganic metal salt. Cathodic electrodeposition (CED), anodic electrodeposition (AED), and electrophoretic deposition (EPD) are the electrochemical synthetic strategies. The cathodic reduction of metal cations corresponds to CED, so MOFs are deposited onto the cathode. In AED, the anode is oxidized producing metal cations for MOF. EPD is a two-step protocol, where MOFs are deposited on the electrode surface in the presence of an applied electrical field. Different parameters can affect the synthesis of MOFs via EPD, such as conductivity, particle size, and so on. The cathodic reduction of metal cations is considered a possible drawback and the use of solvents, such as acrylonitrile, acrylates, and maleates, is strongly suggested for minimizing it.

The ultrasonic (ULS)-assisted approach involves acoustic cavitation phenomena, inducing the increase in temperature and pressure in the reaction system. MOFs at the nanoscale level are obtained with crystallization times faster than those obtained by more conventional approaches.

The mechanochemical method is used to prepare MOFs at room temperature without applying solvents, using a mechanical force for breaking the intramolecular bond in metal salts and creating new bonds between organic ligands and metal cations. On the other hand, this method can be used only for the synthesis of particular MOFs [32].

The reverse microemulsion technique is currently used to synthesize MOFs at the nanoscale level and includes an emulsified liquid phase (water drops in organic solvent or droplets of organic solvent in water). Surfactant molecules act as stabilizers of the emulsified solution and dispersing agents of the resulting nanostructure, thereby increasing MOFs’ stability.

The topology-guided design includes ligands with increasing steric hindrance and changes in synthetic conditions to avoid interpenetration in MOFs. Using this approach, it is possible to tune structure and functionality of MOFs.

Different methods of post synthetic modifications are also reported in the literature [32]. They are not actual synthetic routes, but modification strategies for further functionalizing MOFs’ structures, introducing proper functional groups.

### 2.4. COFs

Two-dimensional covalent organic frameworks (COFs) are porous polymers connecting organic molecules in two dimensions via covalent bonds [34,35]. Their structure and pores size are tunable and characterized by high surface area, thermal stability, and long-lasting porosity.

Several COFs have been designed and synthesized with different sizes, symmetries, and geometries of the structural units, such as COFs based on boric anhydride and borate (BCOFs), on Schiff base (SCOFs), on triazine groups (TCOFs), and on polyimide (PCOFs) [36].

The most common approaches are solvothermal, mechanochemical, solvent-free, microwave-assisted, and sonochemical synthesis [37] (Figure 4).

The solvothermal method is the most common approach, obtaining highly crystalline COFs. As usual, this method is performed at the solvent boiling temperature, also involving high-pressure conditions, as previously reported for TMDs and MOFs (see Section 2.2 and Section 2.3). The main drawbacks are the long-time synthetic procedure and the use of organic solvents.

The mechanochemical synthesis represents an alternative strategy for synthesizing COFs, being easy to perform, efficient, solvent-free, and operating at room temperature. A mechanical force is used for breaking intramolecular bonds in organic molecules, placed in a mortar, and for connecting them in the porous polymeric structure [38]. The addition of small amounts of catalyst in the mortar can accelerate the reaction rate, homogenizing the reagents and thus improving polymer crystallinity [38]. In fact, COFs obtained through this method usually present a lower surface area and poorer crystallinity compared with those prepared by the solvothermal method.

The solvent-free synthesis method is considered an alternative route for COFs, because it is assumed to be environmentally friendly, easy to perform, low-cost, and suitable for large-scale synthesis. The experimental conditions, in terms of temperature and pressure, are comparable with those of the solvothermal approach, involving, in this case, a solid-state catalyst to enhance COFs’ crystallinity and yield. The main disadvantages are due to the use of a solid-state catalyst: high pressure and high temperature.

The microwave-assisted approach has gained increased attention because it is fast and environmentally friendly, involves higher yields, and consumes less energy. Microwave irradiation supports the polymerization process, and different morphologies and sizes, along with faster crystallization, can be achieved by controlling key parameters such as solvency, energy power, reactants’ concentration, and reaction time, as already reported for MOFs in Section 2.3.

The sonochemical method can improve the homogeneity of COFs and accelerate the rate of polymerization and crystallization, due to the significant increase in pressure and temperature induced by the ultrasonic wave.

However, COFs usually have poor electrical conductivity, which is a real disadvantage for their application in electrochemical sensors and biosensors. Combining COFs with different conducting materials, such as carbon-based nanomaterials, metal and metallic oxides nanoparticles, and/or conducting polymers, can be considered an effective strategy to improve COFs’ application in the electrochemical (bio)sensing area.

## 3. Applications of 2D Nanomaterials to Electrochemical (Bio)Sensors for Healthcare and Biomedical Field

Electrochemical (bio)sensors represent a well-known class of diagnostic systems for the healthcare and biomedical field [1,39,40,41]. In fact, they enable the simultaneous analysis of analytes, also at low concentrations and under different conditions, by using simple and low-cost instrumentation. Moreover, portability and miniaturization are possible in many cases [39].

The critical analytical parameters of electrochemical (bio)sensors, such as sensitivity, selectivity, and stability, are mainly controlled by the design of the sensing interface and platform. Different electrochemical techniques are generally involved, such as amperometry (A), chronoamperometry (CA), cyclic voltammetry (CV), linear sweep voltammetry (LSV), differential pulse voltammetry (DPV), square-wave voltammetry (SWV), chronocoulometry (CC), and electrochemical impedance spectroscopy (EIS) [38,39], and their applicability has been improved with the introduction of new functional materials such as 2D nanomaterials [13,42,43,44,45,46]. Several examples of electrochemical (bio)sensors for clinical analysis and healthcare using 2D nanomaterials such as MXenes, TMDs, MOFs, and COFs are reported and discussed below.

### 3.1. Glucose

Diabetes mellitus represents a worldwide public health problem since it is regarded as the seventh cause of death in the world, according to the World Health Organization (WHO) [39,47]. It is a metabolic disorder caused by an altered insulin action involving blood glucose concentration higher than the normal range (4.4–6.6 mol L^−1^). Also, a glucose level decrease represents a pathological condition, called hypoglycaemia, causing a loss of consciousness in the most serious cases. For these reasons, glucose monitoring is fundamental for diabetics, and fast and reliable glucose sensors for healthcare monitoring are required. Considering the great improvements in nanomaterials over the last decade, an increasing interest in nanomaterial-based electrochemical glucose sensors has been highlighted. Combining glucose biosensors with the particular properties of 2D nanomaterials has significantly improved sensitivity and selectivity. Reviews on electrochemical glucose sensors based on 2D nanomaterials have been recently published. However, they are mainly focused on the application of graphene and its derivatives [48,49].

The first example herein reported is a non-enzymatic glucose electrochemical sensor based on a nanocomposite including a Co-based porous metal–organic framework (MOF) such as ZIF-67 and Ag nanoparticles (Ag@ZIF-67) [50]. Poor conductivity is a real disadvantage for most MOFs, because it can imply a decrease in sensitivity. On the other hand, MOFs’ high porosity and flexibility allow the introduction of functional metal nanoparticles (MNPs) within the pores’ cages/channels to form metal@MOF nanocomposites [50], in which MOF acts as a supporter to wrap MNPs, and the incorporated MNPs are uniformly dispersed in the framework. Ag nanoparticles (AgNPs) were selected because of their high conductivity and biocompatibility, and they were encapsulated within ZIF-67 by a sequential deposition–reduction method [50]. In this case, we have to note that, considering the convention that a nanomaterial is only one such if one or more external dimensions range from 1 to 100 nm, ZIF-67 is not a proper nanostructure because it is a dodecahedron with a size of about 200 nm. A glassy carbon electrode (GCE) was then modified with the resulting Ag@ZIF-67 nanocomposite by drop-casting the composite suspension onto the electrode surface. Glucose was analysed using amperometry, and a linearity range from 2 to 1000 µM with a limit of detection (LOD) of 0.66 µM was obtained. Selectivity and stability were evaluated. Selectivity was evaluated in the presence of usual interferences for glucose such as uric acid (UA), ascorbic acid (AA), and acetaminophen (AP) that did not affect the amperometric response. Regarding the operational stability, a slight activity loss of 6.5% was observed after 25 consecutive analysis cycles; considering the long-term stability, a decrease of 5.8% in the electrochemical response was found after one month at room temperature (RT). Reproducibility, repeatability, and application to real samples were not considered.

Another non-enzymatic glucose sensor based on ZIF-67 including Ag-doped TiO_2_ nanoparticles (Ag@TiO_2_NPs) was developed [51]. The hybrid nanocomposite was synthesized using the solvothermal method and drop-casted on GCE. In this case, the poor conductivity of MOF was improved by introducing metal-doped nanoparticles to obtain a nanocomposite, in which MOF acts as a supporter to encapsulate Ag@TiO_2_NPs uniformly dispersed in the ZIF-67 nanostructure. Glucose was amperometrically detected, and a linearity range from 48μM to 1mM with an LOD of 0.99 µM was obtained. Selectivity was evaluated in the presence of UA, AA, AP, and dopamine (DA) as interferences that did not affect the amperometric response. Reproducibility, repeatability, stability, and application to real samples were not examined.

A flexible Ni–Co MOF/Ag/reduced graphene oxide/polyurethane (Ni–Co MOF/Ag/rGO/PU) fiber-based wearable electrochemical sensor was assembled for monitoring glucose in sweat [52]. rGO/PU fiber was produced by wet spinning technology, and a Ni–Co MOF nanosheet was deposited on its surface to set up the Ni–Co MOF/Ag/rGO/PU (NCGP) fiber electrode. Due to Ni–Co MOF’s large specific surface area and high catalytic activity, the fiber sensor showed good electrochemical performances, with a wide linear range of 10 μM–0.66 mM and an LOD of 3.28 μM. In addition, the NCGP fiber electrode displayed significant stretching and bending stability under mechanical deformation. Selectivity and stability were investigated. Selectivity was evaluated in the presence of interferences such as lactic acid (LA), sodium chloride (NaCl), UA, cysteine (Cys), and DA, which did not affect the amperometric response. Long-term stability was evaluated after 7 days at RT, with a 2.90% decrease in the electrochemical response. As the human body temperature can change after exercise, the sensor was also tested at different temperatures, i.e., 24, 30, and 38 °C, and experimental data evidenced that temperature had little influence on glucose detection. An NCGP fiber-based three-electrode system, including an NCGP fiber as a working electrode, Ag/AgCl/rGO/PU as a reference electrode, and a Pt wire as a counter electrode, was sewn with a sweat absorbent cloth and set on a stretchable polydimethylsiloxane film substrate to provide a non-enzymatic sweat glucose wearable sensor, for the real-time monitoring of glucose in human sweat. Data obtained from two different patients were comparable with those coming from a commercial blood glucometer. In addition, a good correlation between glucose level changes in sweat and in blood was found.

A dual-confinement strategy in MOFs’ nanocage-based structure was involved in realizing a sensor for glucose in sweat [53]. The enzyme retained its activity and stability, even if incorporated in an MOF nanocage, thereby avoiding the enzyme leakage and maintaining its conformational structure. Moreover, the mass transport resulted enhanced modifying the MOF-nanocage mesoporosity. ZIF-67 was firstly synthesized through a bottom-up method. Subsequently, GOX and Hemin were uniformly adsorbed on the surface of ZIF-67 (ZIF-67@GOX/Hemin) via coordination between metal cations and the Hemin carbonyl group. ZIF-67@GOX/Hemin acted as a template for the synthesis of a well-defined core-shell composite (ZIF-67@GOX/Hemin@ZIF-8).

The sensor stability was investigated under different conditions, denaturing for the free-enzyme, such as high temperature or treatment with urea, DMF, or DMSO. The sensor retained its bioactivity at 72.1% after exposure at 80 °C, at 84.6% after the urea treatment, at 70.7% after DMF exposure, and at 90.1% after DMSO exposure. Reproducibility was also investigated with satisfactory results in terms of RSD% (2.4%). Fructose, AA, DA, UA, maltose, and lactose were selected as possible interfering molecules, and their presence did not affect the glucose electrochemical signal. Furthermore, a GOX/Hemin@NC-ZIF-based sensor and printed circuit board (PCB) were integrated into a sweatband for the continuous real-time monitoring of glucose in human sweat, and a good correlation between glucose level changes in sweat and in blood was evidenced. The miniaturized, portable and all-integrated glucose sensor included GOX/Hemin@NC-ZIF as a working electrode, carbon paste, and Ag/AgCl paste as a counter electrode and reference electrode, respectively. A linearity range was obtained amperometrically from 50 to 600 μM with LOD of 2 μM.

A flexible carbon cloth (CC) was used as a matrix to grow NiCo_2_O_4_ nanorods using the hydrothermal method, and then to synthesize in situ the composite ZIF-67@GO, including ZIF-67 and graphene oxide (GO). A ZIF-67@GO/NiCo_2_O_4_/CC hybrid system provides a peculiar structure and it was used as a sensing platform for glucose detection [54]. NiCo_2_O_4_ nanorod arrays offer growth sites for ZIF-67 nanocubes and fast electron transport pathways for CC and ZIF-67@GO. The presence of GO in MOF provides high conductivity and stability to the composite ZIF-67@GO. Considering the electrochemical non-enzymatic sensing platform, ZIF-67@GO/NiCo_2_O_4_/CC acted as a working electrode, a platinum plate as a counter electrode, and Ag/AgCl as a reference electrode. After the morphological, compositional, and electrochemical characterization of the working electrode, glucose was detected using chronoamperometry, achieving a linear range from 0.3 μM to 5.407 mM with LOD of 0.16 μM and a response time within 2 s. Selectivity, repeatability, and stability were investigated. AA, UA, DA, and fructose, sucrose, lactose, maltose, as sugars structurally similar to glucose, were selected as possible interfering compounds, and they did not affect the current value even in large amounts. Repeatability was acceptable in terms of RSD% (1.65%). Long-term stability was tested, and the electrochemical response decreased by 4.20% after 30 days, under not better specified experimental storage conditions. Any application to real samples was not addressed.

Bimetallic CuCo-MOFs were synthesized in situ on nickel foam (NF) electrodes through a simple one-pot hydrothermal procedure and the NF-modified electrode was used as a non-enzymatic sensing platform for glucose detection [55]. Ni foam as a supporting material provides a large surface area, stability, and good electrical conductivity, so MOF criticalities of poor stability and electrical conductivity might be solved. A CuCo-MOF/NF electrode, a platinum sheet, and a standard saturated calomel electrode (SCE) acted as working, counter, and reference electrodes, respectively. Chronoamperometry was used to detect glucose, and a linearity range of 0.05–0.5 mM with LOD of 0.023 mM was obtained. AA, UA, DA, and NaCl were identified as possible interferences, and the response was negligible compared with glucose. Reproducibility was addressed with satisfactory results in terms of RSD% (2.17%). Stability and applications to real samples were not considered.

Ni-based metal–organic framework (Ni-MOF) nanosheets were used as precursors to create Ni-MOF@Ni-2,3,6,7,10,11-hexahydroxytriphenylene (HHTP) core@shell structures by introducing HHTP as a π–conjugated molecule [56]. HHTP interacted with free Ni^2+^ ions on a Ni-MOF surface and etched Ni-MOF NSs at the same time. Ni-MOF@Ni-HHTP nanocomposite was drop-casted onto a GCE surface and the modified GC electrode was used to detect glucose using amperometry in an alkaline aqueous solution. Linear concentrations ranging from 0.5 to 2665.5 µM with LOD of 0.0485 µM were obtained. Selectivity, reproducibility, repeatability, stability, and application to real samples were not examined.

In the next example, MOFs are regarded as sacrificial templates for synthesizing hollow metal oxide/carbon architectures with particular properties such as separate inner voids, low density, structural stability, and large specific surface areas [57,58]. Consequently, the mobility of analyte/ions/electrons at the solid/liquid interphase results in accelerated enhancing of the electron transfer from and to the electrode surface. In particular, NiO/Co_3_O_4_/C was developed as hierarchical hollow architecture using Ni-Co MOF as a sacrificial template. The proximity of Ni(II)/Co(II) ions with the organic ligand in the bimetallic MOF structure accelerates the formation of NiO/Co_3_O_4_ nanostructures closely linked to carbon. DFT studies have defined the role of the ingredients of nanocatalysts in the charge-transfer process and their structural and electronic relationship. In fact, the hollow architecture of NiO/Co_3_O_4_/C used both inner and outer surfaces for glucose interaction, while the presence of carbon linked to NiO/Co_3_O_4_ nanostructures enhanced the electron transfer rate. NiO/Co_3_O_4_/C was incorporated into a biodegradable corn-starch bag (BCSB) representing a free-standing, disposable, bendable, low-cost, and fast electrochemical probe for glucose detection.

A scheme of Ni-Co MOF synthesis, a non-enzymatic glucose-sensor-assembling and glucose-detection mechanism is reported in Figure 5.

A linear concentration, ranging from 0.0002 to 10 mM with an LOD of 45 nM, was achieved using amperometry. AP, AA, urea (U), DA, NaCl, citric acid (CiA), UA, and KCl were considered as possible interferences, with the sensor response to interferences resulting as negligible compared with glucose. Long-term stability was tested after 60 days under not better specified experimental storage conditions, and the electrochemical response decreased by 7.90%. The reproducibility and repeatability were investigated with satisfactory results in terms of RSD%, resulting in 2.1% and 2.4%, respectively. BCSB enzyme-free glucose sensor performance was evaluated in human serum real samples, obtaining interesting results in terms of recovery (98.3–102.4%) and RSD (2.19–2.72%).

A microelectrode based on a flexible nanocomposite including carbon fiber (CF) wrapped in rGO-supporting Ni-MOF nanoflake arrays was designed and developed for non-enzymatic glucose detection [59]. Combing the flexibility and conductivity of carbon substrate and highly dense Ni-MOF nanoflake arrays, interesting analytical performances for glucose detection were evidenced for the corresponding Ni-MOF/rGO/CF electrode. In particular, Ni-MOF nanoflake nanoarrays resulted uniformly supported on rGO/CF. The nanocomposite synthesis involved GO wrapping on the CF surface, the electrochemical reduction of GO to rGO, and the growth of Ni-MOF nanoflake nanoarrays on the rGO/CF surface using the solvothermal method. In addition, using conductive rGO/CF-supporting Ni-MOF nanostructures can accelerate the electron transfer from and to the CF electrode. The flexible hybrid CF electrode represented an alternative to more conventional electrode typologies such as GCEs or carbon paste electrodes (CPS) because it can be easily incorporated in miniaturized sensing systems. Glucose detection was performed using amperometry and a linearity range from 6 μM to 2.09 mM with LOD of 0.6 μM was found. KCl, NaCl, U, UA, AA, and DA were investigated as possible interferences and no significant changes to the electrochemical response of glucose were evidenced. Reproducibility was analysed, obtaining an RSD value of less than 5%. Concerning stability, the amperometric signal decreased by 7.40% after five weeks at RT. The sensing system was applied to detect glucose in orange juice samples, obtaining an RSD value of 3.6%.

A N-doped-Co-MOF@polydopamine nanocomposite (N-Co-MOF@PDA), including Ag nanoparticles (N-Co-MOF@PDA-Ag), was synthesized and then applied to assemble electrochemical sensors for glucose non-enzymatic determination [60]. DA polymerizes on the surface of N-Co-MOF, with N-Co-MOF acting as a catalyst. In addition, the N-Co-MOF@PDA nanocomposite was the reducing agent of Ag^+^ ions to Ag nanoparticles with homogeneous size and good dispersion and conductivity. The combination of N-Co-MOFs and AgNPs assured good analytical performance with the non-enzymatic sensor. The N-Co-MOF@PDA-Ag nanocomposite was drop-casted onto a GCE and the resulting modified electrode was used to electrochemically detect glucose via amperometry, obtaining a linear concentration range from 1 μM to 2 mM and LOD of 0.5 μM. Fructose, AA, and UA were used as potential interferences, evidencing a negligible current response relative to glucose. Reproducibility was analysed, obtaining satisfactory results in terms of RSD (4.6%). Stability was investigated evaluating the sensor response after 3600 s from the first injection of glucose solution: the amperometric signal decreased by 3.8%. The sensor was applied to human serum samples, obtaining recoveries ranging from 96% to 110% with RSDs from 3.54% to 4.95%.

A wearable electrochemical sweat sensor including Ni–Co MOF nanosheets deposited on Au/polydimethylsiloxane (PDMS) film was realized for the continuous checking of glucose in sweat [61]. A flexible three-electrode system based on Au/PDMS film was set up by the chemical deposition of a gold layer on PDMS. Then, Ni–Co MOF nanosheets were synthesized by the solvothermal method and deposited on the Au/PDMS electrode surface. A Ni–Co MOF/Au/PDMS (NCAP) electrode was applied to determine glucose amperometrically and showed a wide linear range from 20 μM to 790 μM, with LOD of 4.25 μM. Selectivity and stability were evaluated. Lactose, U, DA, UA, AA, and NaCl were tested as possible interfering molecules that did not affect the electrochemical response of glucose. Long-term stability was tested and the amperometric response decreased by 2.00% after one month, but storage conditions were not indicated. Repeatability and reproducibility were not analysed. Performances under a stretching state were comparable to those under a not-stretching state. Since body temperature changes during sporting activity, as discussed above, the effect of temperature was studied and similar good performances were evidenced at different temperatures. The flexible sensor was then applied to monitor glucose level in volunteers. A sweat-absorbent cloth was used to cover the working area of the sensing system. Changes in glucose level before and after meals were evidenced, and the results were comparable with those obtained in sweat by commercial glucometers. Moreover, changes in sweat glucose concentration were strictly correlated to the values measured in blood.

A skin-attachable and flexible electrochemical biosensor based on ZnO tetrapods (TPs) and MXene (Ti_3_C_2_Tx) was developed for the continuous monitoring of glucose in sweat [62]. Skin-attachable electrodes included a carbon working electrode and were produced by a conventional screen-printing method, using thermoplastic polyurethane (TPU). Glucose oxidase (GOX) was immobilized using glutaraldehyde (GA) as a cross-linking agent on the working electrode modified with a ZnO TPs/MXene nanocomposite, with the last one showing a high specific area and electrical conductivity. A linear concentration range from 0.05 to 0.7 mM with LOD of 17 μM was obtained by chronoamperometry, including a satisfactory mechanical stability (up to 30% stretching) of the template. Reproducibility was acceptable in terms of RSD% (10%). Stability was investigated, and the amperometric response decreased by 10% of its initial value, after 10 days storage at 4 °C. The developed skin-attachable flexible biosensor was used to check glucose levels in sweat before and after a meal, and during physical activity, by continuous in vivo monitoring, and data were correlated with those collected by a conventional amperometric glucometer in blood.

A glucose biosensor was developed based on an electrode surface functionalized with MXene (Ti_3_C_2_Tx), providing binding sites for enzyme immobilization and proper conductivity [63]. Consequently, a transfer channel for electrons produced by the enzymatic redox reaction between GOX and glucose was guaranteed. A linear concentration range from 0.1 to 10 mM with LOD of 12.1μM was obtained via amperometry. UA and AA, tested as interfering compounds, did not affect the electrochemical response of glucose. Repeatability, reproducibility, and stability were not investigated; the biosensor was not applied to real samples.

An electrochemical glucose biosensor was assembled immobilizing GOX onto GCE modified with poly(3,4-ethylenedioxythiophene):4-sulfocalix [4]arene (PEDOT:SCX)/MXene nanocomposite [64]. PEDOT was synthesized by chemical oxidation using SCX as a counter ion and MXene at high temperature under an inert gas atmosphere. Next, a PEDOT:SCX/MXene hybrid film was obtained by the ultrasonication of PEDOT:SCX/MXene (1:1) dispersion. GOX was then immobilized onto chitosan-modified PEDOT:SCX/MXene/GCE. The biosensor was electrochemically characterized and a stable redox peak of FAD-GOX was observed at −0.435 V, showing a direct electron transfer between the enzyme and the electrode surface. A linear concentration range from 0.5 to 8 mM was achieved, with LOD of 0.0225 mM. Selectivity and repeatability were analysed. UA, AA, oxalic acid (OA), L-alanine (L-ala), and L–tyrosine (L-tyr) were tested as possible interferences and no changes were evidenced in the electrochemical signal after the addition of all of them. Repeatability was satisfactory with an RSD of 2.1%. The electrochemical signal decreased by 13% after 20 days at 4 °C. The biosensor was applied to commercial fruit juice samples with satisfactory recoveries, ranging from 96% to 99%.

A scheme of the biosensor assembling is reported in Figure 6.

An advanced butterfly-inspired hybrid epidermal biosensing (bi-HEB) patch is now reported [65]. The bi-HEB patch included a glucose biosensor provided with pH and temperature sensors for precise quantitation and two biopotential electrodes for the real-time monitoring of electrophysiological signals. Nanoporous carbon and MXene (NPC@MXene) nanocomposite including platinum nanoparticles (PtNPs) were used to modify a Au electrode and support GOX immobilization via an EDC/NHS approach for assembling a glucose biosensor. Glucose levels were determined using chronoamperometry and a linearity range of 3 μM–21 mM with LOD of 7 µM was obtained. Reproducibility resulted <5% in terms of RSD. The bi-HEB patch integrated in a wearable system was used to control the sweat glucose and electrophysiological (EP) parameters of human subjects participating in indoor physical activities.

A non-enzymatic sensor based on a nanostructured electrode including MXene, chitosan (CHI), and Cu_2_O nanoparticles was assembled for the simultaneous detection of glucose and cholesterol [66]. MXene and CHI act as a nanostructured sensing platform, while Cu_2_O nanoparticles provide catalytic active edges to improve sensor performances. Glucose and cholesterol were determined simultaneously by CV: a linear concentration range from 52.4 to 2000 μM with LOD of 52.4 μM was achieved for glucose, and a linearity range from 49.8 to 200 μM with LOD of 49.8 μM was found for cholesterol. Sucrose (SC), UA, AP, lactose, NaCl, AA, and L-Cys were tested as possible interferences that did not affect the electrochemical signal of glucose. The sensor was applied to human serum real samples for quantifying glucose and cholesterol simultaneously and acceptable recoveries ranging from 98.04% to 102.94% were evidenced.

A glucose sensor was realized using a GCE modified with MXene/MOFs nanohybrid [67]. The nanocomposite with high conductivity was synthesized by depositing ZIF-67 as an MOF onto two-dimensional Ti_3_C_2_Tx nanosheets as MXene. Ti_3_C_2_Tx nanosheets improved electrical conductivity, while ZIF-67 improved electrocatalytic activity. Glucose was determined amperometrically, and a linearity range from 5 µM to 7.5 mM with LOD of 3.81 µM was obtained. UA, AP, and AA were tested as possible interferences that did not affect the electrochemical response. Repeatability was investigated with satisfactory results in terms of RSD (1.18%). The sensor was not applied to real samples.

An MXene nanocomposite, presenting a combination of MXene with TiO_2_ nanocrystals (Ti_2_C-TiO_2_), was applied to modify GCE for assembling a non-enzymatic glucose sensor [68]. It was synthesized through the oxidation of Ti_2_C-MXene nanosheets. The oxidative opening of the nanosheets produced TiO_2_ nanocrystals on their surface. The combination of MXene nanosheets and TiO_2_ nanocrystals accelerated the electron transfer from and to the sensing surface. The nanocomposite was then casted on the electrode surface and the corresponding modified GCE was used for determining glucose via DPV and CA. A linearity range from 0.1 to 200 µM and LOD of 0.12 µM were evidenced. AA, UA, and DA were selected as possible interferences, evidencing an insignificant electrochemical response compared with glucose. Reproducibility was acceptable in terms of RSD (4%). After 15 days, the signal response decreased by 6.5%, but storage conditions were not indicated. The sensor was applied to human serum samples with recoveries ranging from 99.80% to 100.23%, comparable with those obtained from a commercial glucometer.

A glucose sensor based on a carbon fiber electrode (CFE) modified with cobalt oxide Co_3_O_4_ nanocubes directly grown on a conducting MXene layer was realized [69], where 2D nanosheets deposited on the electrode surface supported the uniform growth of nanocubes. Conductive MXene and Co_3_O_4_ nanocubes as catalytic active sites acted synergistically to support sensor performances. Glucose was determined via amperometry, achieving a linear range from 0.05 μM to 7.44 mM with LOD of 10 nM. AA, DA, AP, catechol, L-Cys, resorcinol, UA, and hydrogen peroxide (H_2_O_2_) were considered as possible interferences that did not affect the glucose response. After 30 days, the signal response decreased by 2.0%, but storage conditions were not indicated. Reproducibility was satisfactory in terms of RSD (2.52%). The sensor was applied to spiked real samples of human serum, urine, and blood, with recoveries from 97.8% to 101.6% with results comparable to those obtained by the conventional colorimetric method.

An MXenes (Ti_3_C_2_T_x_)-based nanocomposite with Cu_2_O nanoparticles (Ti_3_C_2_T_x_-Cu_2_O) was developed to assemble a glucose sensor by modifying GCE [70], as shown in Figure 7. The morphological characterization of Ti_3_C_2_T_x_-Cu_2_O evidenced that the micro-octahedral Cu_2_O nanoparticles were distributed uniformly on the MXene surface.

Glucose was determined using CA and a linear range from 0.01 to 30 mM with an LOD of 2.83 mM was evidenced. AA, NaCl, U, LT, fructose, and SC were considered as interfering molecules, without affecting the sensor response. After 30 days at RT, the analyte response decreased by 5.00%. The sensor was applied to human serum real samples and the data resulted as comparable with those coming from a commercial glucometer.

As a last example, a glucose sensor based on a two-dimensional (2D) conjugated metal–organic framework (c-MOF) film, such as 2D Cu_3_ (HHTP)_2_ (HHTP = 2,3,6,7,10,11- hexahydroxytriphenylene) c-MOF, is reported [71]. The MOF film was deposited onto a Au electrode, improving conductivity and the electrocatalytic response. Glucose was detected amperometrically and a linear range from 0.2 μM to 7 mM with LOD of 10 μM was achieved. NaCl, UA, lactose, U, AA, and DA were tested as possible interferences without affecting the electrochemical response of glucose. Long-term stability was investigated and the amperometric signal remained almost unchanged after 60 days at room temperature. Reproducibility was not considered, and the sensor was not applied to real samples.

As a conclusive comment regarding the reported examples of sensors for the determination of glucose, it can be observed that LOD values are generally micromolar, and nanomolar values were reported for two examples only [57,69].

Notably, the complexity of materials does not always correspond to better performances in terms of linearity range or LOD.

Considering enzyme-based biosensors, only a few examples are reported in the literature [53,62,63,64,66] compared with the total. It is well-known that electrochemical enzyme-based biosensors are easy to assemble and generally reusable, but their major drawback is the stability of the enzyme over time.

Selectivity, applicability to real samples, and sensors’ validation with analytical conventional methods have not always been adequately analyzed and addressed.

The glucose sensors’ analytical performances, together with the corresponding sensor formats, are summarized in Table 1.

### 3.2. Neurotransmitters

Neurotransmitters are endogenous chemical molecules able to send, improve, and exchange specific signals between neurons and other cells.

They can be classified into two groups, considering their electrochemical activity. Electroactive neurotransmitters include dopamine, serotonin, epinephrine, norepinephrine, among others, while glutamate, acetylcholine, and choline are assumed electroinactive [72].

In this review, attention was focused on electrochemical sensors based on 2D nanomaterials for the determination of two important electroactive neurotransmitters such as dopamine and serotonin, where the nanostructured interface acts to amplify the signal and subsequently to improve the performance of the sensor in terms of stability, sensitivity, and sustainability.

#### 3.2.1. Dopamine

Dopamine is a catecholamine neurotransmitter widely present in the central nervous system (CNS). It influences attention skills and brain plasticity, i.e., the ability of neural networks in the brain to change through growth and reorganization. In addition, DA plays a crucial role in memory and learning. Several neurological disorders such as Parkinson’s disease and schizophrenia are associated to abnormal levels of DA [72].

Several studies evidenced that physiological levels of DA in human biological fluids are significantly different. For example, 5 nM DA is reported in urine and cerebrospinal fluid [73], while normal values of DA in blood range from 10 to 480 pM.

The monitoring of DA in the presence of other molecules such as AA, UA, and tyrosine, among others, is essential for the diagnosis of neurological diseases and for understanding mechanisms underlying human neuropathologies.

A Ti_3_C_2_T_x_ (MXene)/Pt nanoparticle-modified GCE (Ti_3_C_2_T_x_/PtNPs/GCE) was developed for determining AA, DA, UA, and APAP [74]. GCE was modified by drop-casting using a dispersion of MXene and PTNPs. MXene acted as a nanostructured sensing platform, while PtNPs provided catalytic active edges to improve the sensor performance. Target analytes were determined using amperometry: the linear response range was observed up to 750 µM for all the analytes, and an LOD of 10 nM was found for DA. In order to make DA detection selective in samples containing AA, an outer layer from an ethanolic solution of Nafion or from chitosan was deposited onto the modified electrode surface. In fact, they did not affect the electrochemical behaviour of DA and UA, their redox behaviour being similar to the one evidenced in the modified GCE without any additional outer layers. Selectivity, reproducibility, repeatability, stability, and application to real samples were not examined.

Another electrochemical sensor for DA was realized using MoS_2_ (TMD) electrodeposited onto pyrolytic graphite sheets (PGSs) and doped with Mn [75]. The presence of Mn as a dopant improved the electrochemical behaviour of DA compared with bare PGS and undoped MoS_2_. In fact, Mn-MoS_2_ enhanced selectivity toward DA in the presence of other common electroactive interferences such as AA and UA. DA was determined through DPV, and two linearity ranges were observed: one was from 50 pM to 5 nM and the other one was from 5 nM to 5 mM, with LOD of 50 pM. Concerning selectivity, the electrochemical response was not affected by the presence of AA and UA. The sensor was tested in artificial samples of human serum and sweat, with LOD values of 50 nM and 5 nM, respectively. Reproducibility, repeatability and stability data were not provided as well as applications to real samples.

A titanium carbide (MXene) (Ti-C-T_x_)-modified GCE was assembled for the simultaneous determination of AA, DA, and UA [76]. It was demonstrated that Ti-C-Tx/GCE evidenced satisfactory electrocatalytic activity and separated oxidation peaks for AA (0.01 V), DA (0.21 V), and UA (0.33 V). Their simultaneous determination was performed at physiological pH, obtaining a linearity range of 100–1000 µM for AA, 0.5–50 µM and 0.5–4 µM for DA, and 100–1500 µM for UA. The LOD values were 4.6 µM, 0.06 µM, and 0.075 µM for AA, DA, and UA, respectively. Reproducibility was investigated involving three different independent sensors, which evidenced no significant changes in the electrochemical response for AA, DA, and UA in PBS. Considering stability, the electrochemical response decreased by 10.4% after 25 days at RT even after repeated use. Selectivity was also tested in the presence of U, nicotine, L-Cys, and APAP, and the electrochemical response was not affected by the presence of interferences. Any application to real samples was not provided.

An electrochemical flexible sensor for DA determination was realized by the direct growth of molybdenum disulphide nanosheets (MoS_2_N_S_) on carbon cloth (CC) [77], as reported in Figure 8.

In MoS_2_N_S_/CC flexible electrodes, CC acts as a conductive and flexible support, and MoS_2_Ns promotes uniform DA adsorption and oxidation. DA was determined by CV, and a linear concentration range from 250 to 4000 μM with LOD of 0.3 μM was evidenced. Repeatability and reproducibility were considered satisfactory in terms of RSD, 1.87% and 1.35%, respectively. The sensor, stored under ambient conditions, was tested against DA every 4 days for 16 days. RSD was 2.9%, indicating good long-term stability. Data concerning real samples application were not provided.

A novel triazine-based covalent organic framework (TS-COF) was synthesized by condensation of 1, 3, 5-tris-(4-aminophenyl) triazine (TAPT) and squaric acid (SA) via a solvothermal approach. Next, AuNPs were grown on a TS-COF surface and rGO was incorporated in AuNPs@TS-COF composite. AuNPs@TS-COF/rGO nanocomposite was then used for modifying GCE and the resulting sensor was used to assay DA, UA, and AA [78]. TS-COF bounded to rGO by π–π stacking promoted electron transfer and electron mobility, while TS-COF acted as a dispersing agent for AuNPs that improved both electrical conductivity and electrocatalytic activity. AA, UA, and DA were determined via DPV, and linearity ranges of 8–900 μM, 25–80 μM, and 20–100 μM were obtained for AA, UA, and DA, respectively, with LODs of 4.30 μM for AA, 0.07 μM for UA, and 0.03 μM for DA. Glucose, L-cys, glycine, CA, tyrosine, lysine, tryptophan, leucine, and L-glutamic acid were used as possible interfering molecules, and they did not affect the determination of AA, DA, and UA, the relative errors being under ± 5%. Reproducibility was considered acceptable in terms of RSD: 4.76% for UA, 3.53% for DA, and 3.15% for AA. Concerning repeatability, satisfactory RSD values were obtained: 2.19%, 4.48%, and 3.26% for AA, DA, and UA, respectively. The sensor was applied to human urine real samples, with recoveries ranging within 97.0% and 104.4%.

A composite was prepared including PMo_12_O_40_^3−^ (PMo_12_) as polyoxmetalate (POM), C_9_H_5_FeO_7_ (MIL-100(Fe) as an Fe-based MOF, and polyvinylpyrrolidone (PVP), and then it was deposited on a GCE for the simultaneous determination of DA and UA [79]. MIL-100(Fe) can encapsulate POMs within its mesoporous architecture. These POM-based MOF composites not only evidenced large surface areas, but their multi-component and multi-interface architectures also improved electron transfer, thereby enhancing analytical performances and the stability of the sensor. PVP can prevent hybrid particles agglomeration, thereby supporting conductivity and increasing electrochemical sensor performance. DA and UA were determined using DPV, and linear concentration ranges of 1–247 μM for DA and 5–406 μM for UA, with corresponding LODs of 1 μM and 5 μM, were obtained. AA, tyrosine, guanosine, isoleucine, alanine, tryptophan, KCl, xanthine, glucose, hypoxanthine, and NaCl were chosen to check selectivity, and no significant response to these interfering molecules was found. Considering long-term stability, DA response decreased by 10.31%, while the one of UA decreased by 11.31%, after 42 days at RT. Repeatability was acceptable in terms of RSD: 2.54% (DA) and 3.18% (UA). Similarly, reproducibility was found to be satisfactory, with RSDs of 2.08% and 5.19% for DA and UA, respectively. The sensor was applied to human serum real samples and recoveries of 97.67–102.16% and 97.81–102.89% for DA and UA, respectively, were found.

An electrochemical sensor for DA determination was realized, modifying a low-cost lead pencil graphitic electrode (LP) by dip-coating using a ZIF-67/PEDOT composite [80], as shown in Figure 9.

ZIF-67 as an MOF has interesting properties such as a crystalline structure, chemical durability, high surface area, high thermal stability, and tunable pore size, while PEDOT improves conductivity and accelerates electrode transfer at the ZIF-67 surface, preserving the peculiar MOF properties such as large surface area and stability. DA was determined using amperometry, and a linearity range from 15 to 240 μM with LOD of 0.04 μM was found. AA, UA, L-Cyst, glucose, and salbutamol were assumed as possible interfering molecules and negligible interfering impact towards DA was evidenced, even in the presence of several-fold higher concentrations of interfering molecules. Reproducibility was acceptable in terms of RSD (0.92%); concerning stability, the amperometric response decreased by 2% after 10 days, but storage conditions were not identified. The sensor was then applied to real blood samples of a COVID-19 quarantine patient, since neurological disorders such as anxiety, depression, and agitation were connected to the lockdown condition, but the results were not accurately explained.

A conductive graphitic pencil electrode (GPE) was modified using a nanocomposite including Ti_3_C_2_Cl_2_ as MXene and 1-methyl imidazolium acetate as ionic liquid (IL) for preparing a DA electrochemical sensor [81]. As discussed in Section 2.1., MXenes are characterized by good conductivity, a high surface area, biocompatibility, hydrophilicity, and resistivity against electrode surface fouling and passivation. However, low flexibility and poor stability in aqueous media and air, due to the presence of hydrophilic functional groups, make those materials easy to oxidize; consequently, their working potential range is limited. For this reason, the stabilization of an MXene surface by IL is an effective strategy to minimize its oxidative degradation without compromising MXene conductivity. An IL/MXene composite was drop-casted onto PGE, and DA was determined via amperometry. A linearity range from 100 μM to 2 mM with LOD of 702 nM was obtained. Repeatability, reproducibility and stability were investigated, and acceptable results in terms of RSD were found: 2.3% (repeatability) and 1.9% (reproducibility), besides an RSD of 1.3% obtained after 14 days storage at RT (stability). UA, AA, glucose, fructose, L-cys, and U were tested as interfering molecules, without affecting the DA amperometric signal. The sensor was applied to spiked real samples of human serum with recoveries ranging from 98.3% to 100.0%.

MOF are considered promising nanomaterials in the sensing area because of their high porosity, adsorption capability, film-forming ability, and tunable synthesis protocol, and nanocomposites including MOF and MXene produce a hybrid with a synergistic combination of the properties of both 2D nanomaterials.

A Ti_3_C_2_ membrane was synthesized by doping UIO-66-NH2 (MOF) with Ti_3_C_2_ (MXene) through a hydrogen bond and used to modify GCE by drop-casting, for assembling a DA electrochemical sensor [82]. DA was determined via DPV, and a linear concentration range from 1 to 250 fM with LOD of 0.81 fM was obtained. Glucose, bovine serum albumin (BSA), AA, and UA were investigated as possible interferences, but they did not affect the DA signal response. After 15 days at 4 °C, the electrochemical response decreased by 7.9%. Reproducibility was found to be good in terms of RSD (3.16%). The sensor was applied to spiked human serum real samples, with recoveries in the range 101.2–103.5%.

A monolayer titanium carbide (Ti_3_C_2_T_x_) material was prepared, and an MXene-based nanocomposite was developed by self-assembling ZnO nanoparticles and a Ti_3_C_2_T_x_ monolayer. It was applied to modify a gold electrode for determining DA [83]. Ti_3_C_2_Tx presents good conductivity, hydrophilicity, a high surface area, biocompatibility, and its integration with another nanomaterial such as ZnO nanoparticles could improve the analytical performances of the sensor. DA was analysed via CA, and a linear concentration range from 0.1 to 1200 μM with LOD of 0.076 μM was evidenced. Glucose, U, serotonin, AA, and UA were investigated as possible interferences that did not interfere with the analysis of DA. Reproducibility and repeatability were considered satisfactory in terms of RSD, at 4.24% and 1.04%, respectively. The sensor was applied to human serum samples, with recoveries ranging from 97.8% to 102.2%.

A hybrid nanomaterial based on Ti_3_C_2_ as MXene, graphitized multi-walled carbon nanotubes (g-MWCNTs), and ZnO nanospheres (ZnO NSPs) was realized for developing an electrochemical sensor for DA assay [84]. Ti_3_C_2_ was combined with g-MWCNTs to enhance stability and electrochemical properties, and the addition of ZnO NSPs could further improve the catalytic and electrochemical properties of the Ti_3_C_2_/g-MWCNTs nanocomposite. Finally, Ti_3_C_2_/g-MWCNTs/ZnO NSPs was deposited on GCE for assembling the DA sensor. DA was determined using DPV, with a linearity range from 0.01 to 30 µM and LOD of 3.2 nM. Glucose, alanine, glycine, leucine, OA, U, AA, and UA were assumed as possible interferences that did not affect the DPV response of DA. Long-term stability was investigated, and a decreased response of 7.4% was evidenced after 25 days at 4 °C. Reproducibility and repeatability were considered acceptable in terms of RSD, 0.54% and 1.16%, respectively. The sensor was applied to human serum samples with recoveries ranging from 98.6% to 105.9%.

A screen-printed electrode (SPCE) modified with Ti_3_C_2_ nanolayers was assembled for the simultaneous determination of DA and tyrosine [85]. SPCE was modified by the drop-casting of MXene suspension. DA and tyrosine were determined using DPV, but only in the linearity range (0.5–600.0 µM,) and LOD (0.15 µM) of DA were reported and commented. The sensor was then applied to determine DA and tyrosine in pharmaceutical drugs and in human urine samples, with recoveries ranging from 97.1% to 104.0% for DA and from 96.7% to 102.5% for tyrosine.

A DA electrochemical sensor based on flower-like MoS_2_ (TMD) nanomaterial, decorated with single Ni site catalyst (Ni-MoS_2_), was realized by modifying GCE [86]. Single atom catalysts (SACs) are a new class of electrocatalysts where isolated metal atoms are distributed and docked on solid supports [87]. SACs have been proposed as single-atom nanozymes (SAzymes) mimicking natural enzymes, due to high catalytic stability, tunable activity, low cost, and high storage stability [88]. MoS_2_ has been investigated as a support material, because of its surface area-to-volume ratio, conductivity, and capability in biomarker detection [89]. Moreover, considering Ni electronegativity and the redox properties of the couple Ni(II)/Ni(III), a flower-like MoS_2_ modified by a Ni atom (Ni-MoS_2_) was developed as a proper and suitable sensing nanohybrid for the selective and sensitive determination of DA. The neurotransmitter was detected using DPV in a linear range from 1 pM to 1 mM, with LOD of 1 pM. AA, UA, glucose, and U were assumed as interfering molecules, and a negligible interfering effect was observed. Repeatability was found to be acceptable in terms of RSD (3.2%), while after 7 days at RT in air, a decreased electrochemical response of 11.4% was evidenced. The sensor was applied to bovine serum samples with recoveries ranging from 97.00% to 105.00%.

MoS_2_ screen-printed electrodes (SPEs) were developed for DA electrochemical detection [90]. MoS_2_ SPEs were developed utilising high viscosity screen printable inks containing MoS_2_ particles in different concentrations and with various sizes, and ethylcellulose as a binder. MoS_2_ inks were printed onto conductive FTO (Fluorine-doped Tin Oxide) substrate. DA was determined via DPV in a linear range up to 300 µM, with LOD of 260 nM. Selectivity, reproducibility, repeatability, stability, and application to the real samples were not examined.

#### 3.2.2. Serotonin

Serotonin is also called 5-hydroxytryptamine (5-HT) and is the most important monoamine neurotransmitter and neuromodulator. Normal values of 5-HT in blood vary in the range 0.6–1.6 µM [91]. It plays a fundamental role in several biological and psychopathological processes, such as sleep regulation, depression, eating disorders, irritable bowel syndrome, anxiety disorders, and psychosis, among others. Consequently, the development of rapid and sensitive methods for detecting 5-HT in body fluids is crucial for supporting the correct diagnosis of neuropathologies and psychopathological disorders.

As a first example, an electrochemical sensor based on GCE modified with a nanocomposite including L-cysteine-terminated triangular silver nanoplates (Tri-AgNPs/L-Cys) and Ti_3_C_2_T_x_ (MXene) nanosheets [92] is reported, and shown in Figure 10. AgNPs are biocompatible and have good electrocatalytic activity but tend to aggregate if any system supporting their dispersion is not used. As the multilayer structure of MXene can prevent nanoparticles’ aggregation, AgNPs can be incorporated between MXene layers, promoting electron transfer and increasing surface area, with improved performances of the sensor as consequence. 5-HT was determined using DPV in a linear range of 0.5–150 μM, with LOD of 0.08 μM. Selectivity was tested using different interfering molecules such as UA, L-Cys, L-Arginine (L-Arg), L-Phenylalanine (L-Phe), and L-Glycine (L-Gly), and little peak current changes were observed. The sensor was used to assay 5-HT in human serum samples, with recoveries ranging from 95.38% to 102.3%.

A flexible sensing platform was developed with a WS_2_/graphene nanostructured hybrid supported on polyimide (WGP) for the simultaneous and selective determination of DA and 5-HT [93]. WS_2_ (TMD) and graphene acted synergistically, producing a nanocomposite with unique electrical conductivity and several electrochemically active sites. DA and 5-HT were detected simultaneously using DPV with linear ranges of 1.21–13.37 µM (DA) and 249 nM–9.9 µM (5-HT) and LODs of 1240 nM (DA) and 240 nM (5-HT). Stability and selectivity were investigated. DA and 5-HT electrochemical responses were recorded every five days over 30 days, with decreases of 16.3% (DA) and 17.5% (5-HT). Storage conditions were not specified. UA, AA, glucose, and epinephrine were identified as possible interferences, and negligible DPV responses occurred for all of them. The sensor was applied to artificial cerebrospinal fluid (CSF) samples because abnormal levels of DA and 5-HT in the CSF are correlated to the malfunctioning of dopaminergic and serotonergic neurons in CNS. Satisfactory recoveries ranging from 95.2% to 104.1% were obtained.

An electrochemical cell-sensing platform for 5-HT detection based on MXene/SWCNTs nanocomposites was assembled onto GCE [94]. The nanohybrid included SWCNTs uniformly distributed on the surface and among the MXene layers. The conductivity and electrocatalytic properties of both the starting nanomaterials improved in the synthesized MXene-SWCNTs composite. 5-HT was determined using amperometry in a linearity range from 4 nM to 103.2 µM with LOD of 1.5 nM. Reproducibility was considered acceptable in terms of RSD (3.8%). In terms of stability, the signal response decreased by 10% after 3 weeks, but storage conditions were not specified. Concerning selectivity, AA, DA, UA, and tyrosine were assumed as possible interfering molecules, yielding negligible current signals. Since the real-time monitoring of 5-HT produced from living cells is crucial for the early diagnosis and rapid treatment of several neuropathogical disorders, the sensor was applied to different cell lines to assess its applicability under physiological conditions.

The same research group had proposed a Ti_3_C_2_T_x_-reduced oxide graphene (Ti_3_C_2_Tx-rGO) nanocomposite for modifying GCE to determine 5-HT in biological fluids [95]. The nanostructured hybrid material provided a large surface area and effective active sites for enhancing sensing layer electrochemical performances and, compared with [94], SWCNTs were substituted by rGO. 5-HT was quantified using DPV, in a linear range of 0.025–147 µM with LOD of 10 nM. Concerning stability, the sensor response decreased by 8.3% after 10 days at 4 °C. DA, AA, and UA were tested as possible interferences, and they did not affect the analyte signal response. Reproducibility was acceptable in terms of RSD (5.2%). The sensor was applied to real samples of human blood plasma and recoveries ranged from 96.4% to 107%.

As a conclusive comment regarding the reported examples of sensors for the detection of neurotransmitters, it can be observed that LOD values are generally micromolar, even if several examples reported lower values ranging from nanomolar to femtomolar [74,75,81,82,84,86,90,93,94,95].

Questionable points concern the validation of the proposed methods. In fact, in all cases, results were not validated with conventional and standard analytical methods.

The data of sensor formats and analytical performances concerning the herein reported examples for the determination of neurotransmitters are summarized in Table 2.

### 3.3. Hormones

According to Starling’s original definition (1905), “a hormone is a substance produced by glands with internal secretion, which serve to carry signals through the blood to target organs” [96]. In other words, hormones are secreted generally by glands, and circulate in blood by simple diffusion before reaching the target cell. Hormones can induce responses of target cells either by interaction with a specific receptor on the cell membrane without entering the cell or by interaction with a receptor within the target cell. Notably, hormones circulate in the blood at very low concentrations (nanomolar amounts or even less); this is the reason why accurate and sensitive approaches are needed for their determination, such as the use of electrochemical sensors integrated with nanomaterials or nanocomposites [97].

Herein, some recent examples of electrochemical sensors based on 2D nanomaterials for hormone determination are reported and discussed.

Human cortisol (11b,17a, 21-Trihydroxypregn-4-ene-3,20-dione) is a steroid hormone, one of the major glucocorticoids synthesized in the zona fasciculata of adrenal glands, and it plays a vital role in emotional responses like stress or depression [98].

A thread-based electrochemical immunosensor was developed for cortisol determination in sweat immobilizing anti-cortisol on L-cys/AuNPs/MXene-modified conductive thread electrode [99], as shown in Figure 11. MXene and AuNPs increase the surface area and promote antibody immobilization, thereby improving sensitivity. The antibody was immobilized on the sensing layer by using EDC and NHS as coupling agents.

Cortisol was analysed using amperometry and a linearity range from 5 to 180 ng mL^−1^ and LOD of 0.54 ng mL^−1^ were observed under optimized conditions. Corticosterone, cortisone, AA, UA, and creatinine were investigated as possible interfering molecules, but their amperometric response did not affect the cortisol response. Considering reproducibility at different cortisol concentrations, the results were acceptable in terms of RSD, ranging from 2.60% to 3.46%. Long-term stability was addressed, and the initial value decreased by 20% after 6 weeks at 4 °C. The sensor was applied to artificial sweat samples, with recoveries in the range 94.47–102%.

A wearable electrochemical impedimetric immunosensor based on a Ti_3_C_2_T_x_-decorated laser-burned graphene (LBG) flake 3D electrode network including a microfluidic channel and chamber was assembled for cortisol determination in human sweat [100]. Polydimethylsiloxane (PDMS) was selected as a substrate for the flexible and stretchable patch sensor. Then, LBG was deposited on it and a laser line gap induced a disconnection between laser-burned graphene flakes and conductivity, so that the electrochemical activity of the LBG electrode was reduced. For this reason, highly conductive Ti_3_C_2_T_x_ was deposited onto the electrode. In addition, this flexible microfluidic system was prepared using 3D-printed mold and PDMS. Finally, anti-cortisol was immobilized on a Ti_3_C_2_T_x_ MXene/LBG/PDMS 3D sensing network and cortisol was determined using EIS. A linear concentration range between 10 pM and 100 nM with LOD of 3.88 pM was achieved. Aldosterone, corticosterone, prednisolone, and progesterone were tested as possible interferences that did not significantly affect the response of cortisol. Reproducibility with an RSD of 4.6% was considered satisfactory. The wearable sensor was applied to quantify the cortisol in freshly collected spiked sweat samples to evaluate its applicability in clinical diagnostics, and recoveries ranging from 93.68% to 99.1% were obtained.

Epinephrine (EP) or adrenaline is a hormone belonging to the catecholamine family secreted by the suprarenal gland. Catecholamines stimulate CNS and cardiac contraction [101]. Adrenaline is present in human brain fluid, blood, and body fluids at nanomolar levels. Alterations in EP levels can be correlated to several disorders, such as Alzheimer’s or Parkinson’s disease, and multiple sclerosis.

A nanocomposite including reduced graphene oxide (rGO) and Ti_3_C_2_T_x_ (GMA) was used to modify an indium tin oxide (ITO) electrode for determining EP [102]. Ti_3_C_2_T_x_ was uniformly deposited on the 3D G layer and the 3D structure of the GMA supported the interaction with biomolecules. EP was determined using DPV, and a linear concentration range of 1–60 μM with LOD of 3.5 nM was observed. UA was considered a possible interfering compound and any appreciable changes in analytical parameters were evidenced. Repeatability was acceptable in terms of RSD (1%); concerning stability, the voltammetric response retained 96.2% of its initial value after 16 days at RT. The GMA sensor was applied to human urine real samples for verifying its application in clinical analysis and recoveries ranging from 95.7% to 105.7% were evidenced.

An electrochemical sensor based on GCE modified by CoMn-based porous MOF deposited on carbon nanofiber (CNF) was prepared for EP determination [103]. CoMn-ZIF/CNF combined CNF’s good conductivity and CoMn-ZIF’s electrocatalytic properties. EP was determined using DPV, and a linear range from 5 to 1000 µM with LOD of 1.667 µM was achieved. Reproducibility was satisfactory in terms of RSD (1.95%); the voltammetric response retained 87.81% of its initial value after 20 days, under not specified storage conditions. Different amino acids such as tyrosine and leucine, UA, DA, norepinephrine, and glucose and drugs such as ibuprofen and amoxicillin were selected as possible interferences, and they exhibited negligible influence on the EP electrochemical signal. The sensor was applied to human urine samples for evaluating its applicability to a real matrix, with recoveries from 97.51% to 102.53%.

Insulin is an important hormone secreted by β-cells in the pancreas. This hormone regulates the glucose level in the blood through the control of the metabolism of proteins, carbohydrates, and lipids. The normal value of insulin concentration in blood is 50 pmol L^−1^ [104]. Nevertheless, inflammatory autoimmune and/or metabolic disorders can undermine β-cells inducing type I (insulin-dependent) diabetes.

An aptasensor for insulin assay in human serum was assembled modifying disposable Au electrodes (DGEs) with a nanocomposite including copper (II) benzene-1,3,5-tricarboxylate (Cu-BTC) nanowires and a leaf-like zeolitic imidazolate framework (ZIF-L) as MOF [105], as illustrated in Figure 12.

Aptamers were immobilized via physical adsorption on a Cu-BTC/ZIF-L composite that enhanced the aptasensor electrochemical performance due to the integration of electrocatalytic properties and the larger surface area of MOF and Cu-BTC nanowires. Insulin was analysed using DPV, obtaining a linear concentration range from 0.1 pM to 5 μM with LOD of 0.027 pM. Glucose, DA, AA, and melatonin (MELA) were used as interferences for evaluating aptasensor selectivity: the error percentage of 10.5% evidenced the good selectivity of the sensor. Reproducibility was investigated and acceptable results in terms of RSD (5.5%) were obtained. Regarding operative stability, the voltammetric signal decreased a little after continuous measurements up to 20 cycles, evidencing a stability rate of 91.7%. The aptasensor was applied to human serum samples with recoveries ranging from 97.2% to 98.5%, and the data were comparable to those coming from the ELISA standard method. The aptasensor was used to monitor in vivo insulin in non-diabetic and diabetic mice, and data were comparable to those coming from the ELISA standard method, as shown in Figure 13. Biofouling was observed due to the presence of proteins, so the use of anti-biological attachment polymers, such as polyethylene glycol and polyhydroxy ethylmethacrylate, was suggested to prevent biofouling and to improve aptasensor durability.

As a conclusive comment on the reported sensors for the detection of hormones, linearity ranges were sufficiently wide, with LOD values being generally nanomolar as well as picomolar, independently from analyte, and according to hormonal levels in biological fluids such as blood, urine, or sweat.

Notably, an example involving 2D-based biosensor nanomaterials for biomedical applications was used for the in vivo determination of mice for the first time, with important results for the possible commercialization of the device [105].

Analytical performances of the electrochemical sensors reported for the determination of hormones as well as the corresponding sensor format are summarized in Table 3.

### 3.4. Pathogens

A pathogen is defined as an organism causing disease to its host, and virulence is the severity of the disease symptoms [106]. Pathogens are biologically different organisms and comprise viruses and bacteria as well as unicellular and multicellular eukaryotes. In this review, examples of (bio)sensors based on 2D nanomaterials for the assay of bacteria and viruses are reported.

#### 3.4.1. Bacteria

Bacteria are the most common cause of foodborne diseases and present different shapes, types, and properties. Pathogenic bacteria directly or indirectly infect food and water sources, and the ingestion of contaminated foods can induce intestinal infectious diseases and/or food poisoning [107].

In this review, recent examples of electrochemical (bio)sensors based on 2D nanomaterials for the determination of *Escherichia coli (E. coli)*, *Salmonella Mycobacterium tuberculosis*, and two vibrio bacteria such as *Vibrio vulnificus* (VV) and *Vibrio parahaemolyticus* (VP) have been considered.

The first two examples concern the determination of Gram-negative bacterium *E. coli*, a facultative anaerobic rod present in the intestinal tract of animals and humans from birth. A wide and differentiated class of bacteria includes *E. coli*: most strains are not pathogenic while a few induce diseases in humans and animals.

An electrochemical immunosensor was developed based on magnetic COF [108]. A specific egg yolk antibody (IgY) with affinity for *E. coli* was labelled with a porous magnetic covalent organic framework (m-COF) microbeads to assemble a capture probe (m-COF@IgY) for efficiently recognizing *E. coli*, as shown in Figure 14.

m-COF@IgY was then used with ferrocene boronic acid (FBA) as signal tag to combine with *E. coli* in a sandwich complex, dropped on a screen-printed electrode (SPE). The voltammetric signal generated by FBA in the sandwich complex was used for the quantitation of bacterium. After experimental conditions had been optimized, *E. coli* was determined by SVW in the linear range of 10–10^8^ CFU mL^−1^ with LOD of 3 CFU mL^−1^. Immunosensor specificity was investigated using *Vibrio parahaemolyticus* (VP), *Salmonella typhimurium* (ST), and *Listeria monocytogenes* and even at concentration 10 times higher than *E. coli*, the sensor had no significant response to them compared with that from *E. coli*. SPE maintained 97.0% of the initial signal response after being reused 60 times for various samples by controlling the sandwich complex through the magnet, so that the sensing platform could be considered reusable. Long-term stability was also addressed, and the voltammetric signal decreased by less than 10% after 3 months at 4 °C. The sensor was applied to determine *E. coli* in several spiked food samples such as milk, beef, and shrimps, with recoveries ranging from 90% to 103%. The results were validated and compared with those obtained by the ELISA standard method.

A cationic covalent organic polymer (COP), named CATN, was employed to develop an impedimetric sensor for *E. coli* cells [109]. COPs are a class of porous organic materials, generally including polycyclic aromatic hydrocarbons linked by C-C σ bonds and several π electronic systems, thereby guaranteeing good thermal and chemical stability and representing a proper sensing platform [110]. CATN was deposited via electrophoresis onto the interdigitated electrode array (IDEA) and EIS response showed a linear logarithmic relationship with increasing concentrations of *E. coli* up to 10 CFU mL^−1^, with LODs of 2 CFU mL^−1^. Repeatability, reproducibility and stability were not analysed. The sensor was not applied to real samples.

A two-dimensional porphyrin-based covalent organic framework (Tph-TDC-COF) was used for developing an electrochemical aptasensor to determine *E. coli* [111]. Tph-TDC-COF was synthesized starting from 5,10,15,20-tetrakis(4-aminophenyl)-21H, 23H-porphine (Tph), and [2,2′-bithiophene]-2,5′-dicarbaldehyde (TDC) and a highly conjugated structure was obtained that had high conductivity, a large specific surface area, and was ideal for immobilizing aptamers through π-π stacking, hydrogen bonding, and electrostatic interactions. In particular, the specific recognition between the aptamer and *E. coli* results in the formation of the G-quadruplex. Tph-TDC-COF was deposited on AuE and *E. coli* was determined using EIS and DPV. The same linearity range was obtained using EIS and DPV (10–10^8^ CFU mL^−1^), while two different LOD values were found: 0.17 CFU mL^−1^ (EIS) and 0.38 CFU mL^−1^ (DPV). *Staphylococcus aureus*, *Basophils* (Bas), *Staaue* (Sta), and *Salmonella typhimurium* (ST) were considered as possible interferences: the EIS response obtained by a mixture of interferents and *E. coli* was about 8% higher than the EIS signal obtained by detecting *E. coli* alone. Reproducibility was acceptable in terms of RSD (4.92%). The electrochemical response remained almost stable after 15 days at 4 °C. The aptasensor was applied to spiked real samples of bread and milk, with recoveries ranging from 100.09 to 103.97% (bread) and from 99.61 to 100.71% (milk).

*Salmonella* is a flagellated Gram-negative, non-spore-forming bacillus, growing at temperatures between 35 °C and 37 °C. It is a foodborne pathogen because most infections are due to the ingestion of contaminated food. *Salmonella* induces salmonellosis, the most significant symptoms of which are nausea, vomiting, abdominal pain, and diarrhoea, among others.

An MXene/poly (pyrrole) (PPy)-based bacteria-imprinted polymer (MPBIP) sensor was assembled to modify GCE for determining *Salmonella* [112]. MPBIP was prepared via the one-step electropolymerization of pyrrole, and a *Salmonella* template was then eluted. The interaction of *Salmonella* surface groups with MXene functional groups enhanced the connection between MPBIP and the target, assisting the biorecognition process. A linear relationship with the logarithmic concentration of *Salmonella* was found from 10^3^ to 10^7^ CFU mL^−1^ and the corresponding LOD was 23 CFU mL^−1^. Repeatability was acceptable in terms of RSD (0.91%), and the sensor maintained 94.5% of its initial response after 7 days at 4 °C. *Staphilococcus aureus*, *E. coli*, and *Lysteria monocytogenes* were tested at the same concentration of *Salmonella* for evaluating specificity and selectivity, without any significant response compared to that from *Salmonella*. The impedimetric sensor was applied to quantify *Salmonella* in drinking water samples, with recoveries in the range 96–109.4%.

*Mycobacterium tuberculosis* (*M. tb*) is a human pathogen that causes tuberculosis (TB), a serious infectious disease among the top 10 causes of death worldwide.

A genosensor was assembled for the determination of a DNA target inside an IS6110 sequence of the *M. tb* genome, using MXene (Ti_3_C_2_) nanosheets and PPY as modifiers of GCE, ssDNA as Probe DNA (p IS6110), and methylene blue (MB) as a redox indicator [113]. ssDNA was immobilized onto MXene/PPY/GCE via covalent bonding. The combination of MXene and PPY improved the electron transfer rate, conductivity, and sensitivity of the genosensing platform. The analytical performances of the genosensor were evaluated by recording the MB electrochemical response as a function of DNA target concentration using DPV. The MB signal intensity decreased as the DNA target concentration increased, since the redox probe electron transfer decreased because of the hybridization among the ssDNA immobilized onto the electrode and the target DNA. Under optimized conditions, a linearity range of 100 fM–25 nM and LOD of 11.24 fM were observed. Two-base mismatch DNA (2 m-DNA), five-base mismatch DNA (5 m-DNA), complementary target DNA, non-complementary DNA (nc-DNA), and, also, the genomic single-strand of other bacteria similar in sequence to *M. tb*., including the *M. bovis* BCG strain GL2 and *M. simiae* strain TMC 1226, were tested for evaluating specificity. It was evidenced that the MB current intensity increased with the number of mismatches. Reproducibility was acceptable in terms of RSD (6.05%). The biosensor applicability was investigated by analysing *M. tb*-extracted DNA from clinical patient sputum samples. The same samples were also analysed using a conventional PCR method. Recoveries were in the range of 90.52–100.8% and results were comparable with those coming from the PCR method.

The genus *Vibrio*, belonging to the family *Vibrionaceae*, includes more than 35 species, and more than one-third are pathogenic to humans [114]. *Vibrios* are Gram-negative straight or curved rods. Among the pathogenic *Vibrios*, *Vibrio cholerae*, *Vibrio parahaemolyticus*, and *Vibrio vulnificus* have to be mentioned.

As one of the most dangerous *Vibrios*, *Vibrio vulnificus* (VV) could cause acute gastroenteritis, primary sepsis, necrotizing wound infection, and even death.

A dual-mode immunoassay based on electrochemiluminescence (ECL) coupled with Surface Enhanced Raman Spectroscopy (SERS) was designed for the determination of VV [115]. A multifunctional sensing platform (R6G-Ti_3_C_2_T_x_ @AuNRs-Ab_2_/ABEI), including Ti_3_C_2_T_x_ MXene, Rhodamine 6G (R6G), gold nanorods (AuNRs), detection antibodies (Ab_2_), and N-(4-aminobutyl)-N-ethylisoluminol (ABEI), was developed as a signal unit. The dual-mode immunosensor included a capture unit, i.e., Fe_3_O_4_@Ab_1_ where Fe_3_O_4_ supported the immunosensor assembling and Ab_1_ acted as capture antibody. The signal unit R6G-Ti_3_C_2_T_x_ @AuNRs-Ab_2_/ABEI evidenced good conductivity and large specific area. AuNRs and SERS signal R6G molecules were deposited on the surface of Ti_3_C_2_T_x_ by electrostatic adsorption, while R6G, detection antibody Ab_2_ and ECL signal tags ABEI were immobilized onto AuNRs. After capture unit Fe_3_O_4_@Ab_1_ was assembled onto a magnetic glassy carbon electrode (m-GCE), VV was recognized and captured by Fe_3_O_4_@Ab_1_, followed by the formation of the capture unit-target-signal unit immunocomplex Fe_3_O_4_@Ab_1_-VV-R6G-Ti_3_C_2_T_x_@AuNRs-Ab_2_/ABEI after immobilizing the signal unit R6GTi_3_C_2_T_x_@AuNRs-Ab2/ABEI. Consequently, ABEI produced an ECL signal and R6G produced a SERS signal separately. Under optimized conditions, ECL intensity increased with VV concentration. A linear relationship between ECL and the logarithm of VV concentration was found in the range 1–10^8^ CFU mL^−1^ with limit of quantitation of 1 CFU mL^−1^. Considering SERS determination, there was a linear relationship between SERS intensity and the logarithm of VV concentration in the range 10^2^–10^8^ CFU mL^−1^ with a limit of quantitation of 10^2^ CFU mL^−1^. Considering operational stability, ECL response was stable after 16 consecutive scan cycles, with an RSD of 2.0%. In addition, SERS intensity was stable after 10 consecutive measurements, with an RSD of 2.9%. Reproducibility was also investigated, with acceptable results in terms of RSD: 3% for ECL determination and 2.9% for SERS determination. *Vibrio parahaemolyticus* (VP), *Shewanella marisflavi* (SM), *Vibrio harveyi* (VH), and *Enterobacter cloacae* (EC) were analysed as common interfering bacteria in a mixture with VV, providing signal intensities roughly similar to those of blank. The immunosensor was applied to VV-spiked real samples of seawater with recoveries ranging from 94.8% to 110.3% for ECL, and from 92.3% to 112.1% for SERS.

The consumption of raw or undercooked seafood such as crabs, shrimps, scallops, seaweed, oysters, and clams can induce gastroenteritis due to the presence of *Vibrio parahaemolyticus* (VP). In severe cases, the bacterium can cause dysentery, primary septicaemia, or cholera-like illness with the possibility of death [114].

A dual-mode electrochemical and colorimetric aptasensor was developed for the on-site detection of (VP) in shrimps [116]. Mercapto-phenylboronic acid (PBA), ferrocene (Fc), and Pt nanoparticles were assembled on an MXenes layer to develop the nanoprobe PBAFc@Pt@MXenes that acted as dual-signal probe, evidencing peroxidase mimic features and electrochemical activity. A screen-printed Au electrode functionalized with an aptamer acted as a capture probe for VP. A sandwich complex was produced after the conjugation of PBAFc@Pt@MXenes with the capture probe. The colorimetric determination of VP was related to the chromogenic reaction of tetramethylbenzidine (TMB)-H_2_O_2_, catalysed by PBA on the probe, while the electrochemical response of Fc on the probe was further used for the determination of VP. Under optimal conditions, VP was determined electrochemically using DPV with a linearity range of 10^1^–10^8^ CFU mL^−1^ and LOD of 5 CFU mL^−1^, while a linearity range of 10^2^–10^8^ CFU mL^−1^ with LOD of 30 CFU mL^−1^ was obtained by the colorimetric method. *Staphylococcus aureus*, *Salmonella typhimurium*, *Listeria monocytogenes*, and *E. coli* were analysed as common interfering bacteria in a mixture with VP, the signal intensities resulting as roughly similar to those of blank. The electrochemical signal retained 89.6% of the initial value after 2 weeks under not clearly specified storage conditions. The aptasensor was applied to real samples of shrimps with recoveries ranging from 95.0% to 104.3%, in good agreement with those obtained by the colorimetric mode (94.2% to 102.5%).

#### 3.4.2. Viruses

Viruses are everywhere and can induce life-threatening diseases in humans and animals. Indeed, it is well-known that viral infections cause a third of deaths worldwide [117,118].

Virus biosensors have been described in recent reviews, also providing a comparison with the conventional analytical methods [118,119,120,121]. In this review, attention was focused on electrochemical biosensors based on 2D nanomaterials, considering the literature of the last five years (2018–2022) and integrating the data of a previous review [117].

Almost four years ago, the worldwide coronavirus 2019 (CoV-2) pandemic was announced. It is well known that the CoV-2 virus induces Severe Acute Respiratory Coronavirus Syndrome SARS-CoV-2 [118]. Coronaviruses are enveloped viruses, and they can infect humans and animals. COVID-19 contains four structural proteins and protein S on the virus surface is responsible for infection transmission.

The first three examples include MOF-based biosensors but, notably, there is an interesting review focused on the use of MOF for the determination of viruses [122] comprising the literature up to 2019.

An electrochemical dual-aptamer biosensor based on NH_2_-MIL-53(Al) as MOF, Au@Pt nanoparticles, horseradish peroxidase (HRP), and hemin/Gquadruplex DNAzyme (GQH DNAzyme) as a signal nanoprobe was developed for the detection of SARS-CoV-2 nucleocapsid protein (NP) [123]. Au@Pt/NH_2_-MIL-53 was modified with HRP and with the thiolated aptamers (SH-2G-N48 and SH-2G-N61) including a double G-quadruplex sequence, for amplifying the aptasensor response and for catalyzing the oxidation of hydroquinone (HQ). In the presence of NP, HQ was determined using DPV, obtaining a linear correlation with the logarithmic concentration in the range of 0.025–50 ng mL^−1^ with LOD of 8.33 pg mL^−1^. Several proteins, such as cTnI, Her2, and MPT64, were selected as interferences, and they did not affect the SARS-CoV-2 NP response. Repeatability was analyzed with acceptable results in terms of RSD%, ranging from 2.6% to 5.0. Results were comparable to those coming from the ELISA method, and recoveries were in the range 92.0–110%. A scheme of the nanoprobe’s assembling and sensing mechanism are shown in Figure 15.

A label-free electrochemical immunosensor for the determination of the SARS-CoV-2 S-protein [124] was assembled modifying SPCE with SiO_2_@UiO-66 nanocomposite, including UiO-66 and a Zr-MOF nanostructure. The nanocomposite showed high surface area and porosity, good thermal conductivity, and chemical stability. SiO_2_ nanoparticles improved the electron transfer and the conductivity of Zr-MOF. Angiotensin-converting enzyme 2 (ACE2) has been used as receptor for the S-protein [124]. S-protein determination was performed by EIS, and a linear concentration range from 100.0 fg mL^−1^ to 10.0 ng mL^−1^ with LOD of 100.0 fg mL^−1^ was obtained. Human coronavirus HCOV, l-glucose, l-Cys, l-Arg, UA, DA, AA, vitamin D, ribavirin, zanamivir, favipiravir, remdesiver, and tenofovir were selected as possible interferences, and they did not affect the determination of S-proteins, except favipiravir, remdesiver, and tenofovir, since they are antiviral drugs. Reproducibility and repeatability were acceptable in terms of RSD (4.85%). The immunosensor was considered reusable, [124] and was applied to nasal fluid samples, with satisfactory recoveries ranging from 91.6% to 93.2%. Results were validated with the PCR test.

An aptasensor was developed using an aptamer and an imprinting polymer (MIP) for the determination of an intact SARS-CoV-2 virus [125]. The aptasensor was based on SPCE modified with nickel-benzene tricarboxylic acid-MOF (Ni_3_(BTC)_2_ MOF), SARSCoV-2 S-protein aptamer, and polydopamine synthesized via electropolymerization (ePDA). MIP synthesis was performed via PDA electropolymerization on the aptamer [SARS-CoV-2 virus] complex, immobilized on the modified SPCE. The template virus was removed just after the electropolymerization ended. Analytical performances of the MIP-aptamer nanohybrid sensor were evaluated by measuring EIS response to different concentrations of SARS-CoV-2 virus, and under optimized experimental conditions, a linear relationship with logarithmic concentration in the range of 10–10^8^ PFU mL^−1^ with LOD of 3.3 PFU mL 1 was achieved. SARS-CoV, MERS-CoV, influenza A H1N1, and influenza A H3N2 were selected as possible interfering viruses, and no significant response was observed for all of them. Reproducibility and repeatability were considered satisfactory in terms of RSD, at 4.2% and 1.4%, respectively. Considering long-term stability, EIS response did not show significant changes after 14 days at 4 °C. The aptasensor was applied to real saliva and nasopharyngeal swab samples in a viral transport medium (VTM) of sick and healthy patients for the qualitative and quantitative analysis of the virus. Results in both cases were comparable to those obtained via PCR, with recoveries ranging from 98% to 104%.

The last example includes an MXenes-based genosensor, reminding that a recent review focusing on the use of MXenes for the determination of the SARSCoV-2 virus has been reported in the literature [126].

A Ti_3_C_2_T_x_ MXene was functionalized with a single-stranded DNA (ssDNA) through noncovalent adsorption, which facilitates the sequence-specific detection via hybridization with a target SARS-CoV-2 gene [127]. Consequently, ssDNA/Ti_3_C_2_T_x_ was used for developing a chemoresistive biosensing platform for the determination of the SARS-CoV-2 N gene. The hybridization of the SARS-CoV-2 N gene with complementary DNA probes induced the detachment of dsDNA from the Ti_3_C_2_T_x_ layer, so an increase in Ti_3_C_2_T_x_ conductivity occurred. LOD below 10^5^ copies mL^−1^ in saliva was obtained, with a linear concentration range from 10^5^ to 10^9^ copies mL^−1^. Reproducibility, repeatability, selectivity, and stability data were not provided.

The hepatitis B (HBV) virus is widely spread worldwide and is transmitted through blood and body fluids. HBV belongs to the *Hepadnavirus* family; it is an enveloped icosahedral DNA virus with a spherical shape [118]. The virus’s outer layer is a denominated surface antigen (HBS Ag) and it is a lipid envelope containing viral proteins responsible for the host cells attack.

An electrochemical genosensor, based on a modified GCE, was prepared for detecting HBV DNA by combining the electroactive Cu-MOF as signal nanoprobe and electroreduced GO (ErGO) as signal amplification material [128], without any enzymes, labels, or other redox indicators. Cu-MOF with a strong π-conjugate system can interact with ErGO through π–π interaction, enhancing the analytical performances of the sensor. Moreover, the strong interaction between Cu-MOF and DNA through covalent bonding improves stability. The genosensor was used to detect HBV using DPV, and a linear concentration range between 50.0 fM and 10.0 nM with LOD of 5.2 fM was evidenced. Reproducibility was evaluated satisfactory with RSD 3.02%. The voltammetric signal decreased by 4.5% after 2 weeks at 4 °C. The genosensor was applied to spiked human serum and urine samples, obtaining recoveries from 95.2% to 99.8%.

An electrochemical immunosensor based on GCE modified with Cu-MOF was developed for HBS Ag detection [129]. In particular, amine-functionalized Cu-MOF nanospheres were synthesized, for immobilizing Ab via covalent interaction between the Ab carboxyl group and the Cu-MOF amino groups. In addition, the nanospheres acted as electrocatalysts and electrochemical signal amplifiers. After the optimization of the experimental conditions, HBS Ag was determined by means of DPV and a linearity range from 1 ng∙mL^−1^ to 500 ng∙mL^−1^ with LOD of 730 pg∙mL^−1^ was found. Considering reproducibility, RSD was 3.24%. Selectivity was tested by comparing the electrochemical response of the target antigen with those coming from other hepatitis virus biomarkers such as HAV, HDV, and HCV. HBS Ag, and the electrochemical response was much higher compared with other hepatitis virus markers. The biosensor was applied to spiked clinical samples, obtaining recoveries from 79.63% to 92.18%.

As a final remark regarding the biosensors for pathogen analysis herein reported, LOD values achieved fM or fg mL^−1^ in different cases. Concerning the biosensor format, the number of immunosensors, genosensors, and aptasensors is very similar.

Analytical data are generally sufficiently accurate, evidencing, in many cases, applicability to real samples. In some cases, results obtained from real samples were validated with external or standard methods.

Analytical performance and sensor format of the reported electrochemical (bio)sensors for pathogens determination are summarized in Table 4.

### 3.5. Cancer Biomarkers

According to the literature [130], biomarkers can be defined as “a characteristic that is objectively measured and evaluated as an indicator of normal biological processes, pathogenic processes, or pharmacologic responses to a therapeutic intervention”. In particular, cancer biomarkers (CBs) are specific proteins and/or oligonucleotides spread into body fluids during the early stages of cancer at abnormal levels with respect to those found in healthy people. Moreover, they are important in providing data indicating the type and phase of the cancer and to monitor and evaluate treatment efficacy [131].

Herein, recent examples of electrochemical biosensors based on 2D nanomaterials for the determination of some relevant CBs are reported.

Carcinoembryonic antigen (CEA) is a cell membrane structural protein produced by embryonic gastrointestinal tissue and epithelial tumours and is one of the most important clinical CBs for the diagnosis of colon and breast cancers, ovarian carcinoma, colorectal cancer, and cystadenocarcinoma. Its normal level should be below 5 ng mL^−1^ in human serum. Concentration of 5 ng mL^−1^ is assumed as a threshold for differentiating abnormal from normal expression. In fact, CEA concentration is higher than 20 ng mL^−1^ in cancer patients [132].

An integrated microfluidic electrochemical platform was organized for the determination of CEA [133]. It included two functional parts: a herringbone-embedded microfluidic chip and an electrochemical aptasensor. The electrochemical aptasensor was based on SPCE modified with nanocomposite including MXene (Ti_3_C_2_) nanosheets and functionalized carbon nanotubes (CCNTs) with hemin for electrochemical signal amplification. Hemin supported the aptamer immobilization via EDC-NHS covalent coupling. The herringbone-embedded chip produced local mixed flow and enhanced the interaction between CEA and the sensing interface. All the analytical processes involving sample injection, efficient enrichment, target capture, and detection was performed at one integrated platform. CEA was determined using DPV and a dynamic concentrations range of 10–1 × 10^6^ pg mL^−1^ with LOD of 2.88 pg mL^−1^ was obtained. Human serum albumin (HSA), immunoglobulin G (IgG), and glucose were tested in mixture with CEA as possible interfering molecules, and no significant response was observed for all of them. Reproducibility was considered satisfactory in terms of RSD (3.6%). The integrated platform was applied to spiked human serum samples, obtaining acceptable recoveries in the range of 95.29–105.19%.

A label-free immunosensor for CEA assay was realized including β-cyclodextrin (β-CDs) and gold nanoparticles (AuNPs) (Au-β-CD) deposited on the surface of FTO modified with a composite consisting of PANI and MXene (MXene@PANI) [134]. MXene@PANI improved the electrocatalytic activity and conductivity of the immunosensing platform, while Au-β-CD supported anti-CEA immobilization, as shown in Figure 16.

The resulting BSA/anti-CEA/Au-β-CD/MXene@PANI/FTO immunosensor was electrochemically and morphologically characterized and, under optimized conditions, CEA was determined using DPV. The electrochemical signal decreased as the concentration of CEA increased, because the electron transfer was hindered after immunoreaction between anti-CEA, immobilized on the sensing interface, and CEA. There was a linear relationship between current intensity and the logarithm of CEA concentrations in the range 0.5~350 ng mL^−1^ with LOD of 0.0429 ng mL^−1^. Reproducibility was considered satisfactory in terms of RSD (3.61%). AA, glucose, BSA, human immunoglobulin G (IgG), and cancer antigen 15-3 (CA15-3) were selected as possible interferences, without affecting the CEA DPV response. Concerning long-term stability, the electrochemical response decreased by only 9.4% after 10 days at 4 °C. The applicability of the CEA immunosensor was investigated by analysing several spiked real samples of human serum and recoveries ranging from 97.52% to 103.98% were obtained.

A nanocomposite including trimetallic nanoparticles (Au-Pd-Pt NPs) and MXene (Ti_3_C_2_T_x_) nanosheets was drop-casted onto a GCE for assembling a CEA aptasensor [135]. In particular, CEA detection was performed via exonuclease III (Exo III)-supported recycling amplifications using triple-helix complex probes (THC). A CEA target interacted with the aptamer containing hairpin probes to cause cyclic cleavage of the secondary hairpins supported by Exo III to release ssDNAs. The obtained ssDNA hybridized with G-quadruplex-integrated triple-helix complex (THC) signal probes onto the electrode. Then, Exo III cyclically cleaved dsDNAs to release G-quadruplex sequences able to constrain hemin on the electrodic surface. Then, hemin catalysed H_2_O_2_ reduction at Au-Pd-Pt/Ti_3_C_2_T_x_/GCE thus enhancing CEA electrochemical response. A linearity range from 1 fg mL^−1^ to 1 ng mL^−1^ with LOD of 0.32 fg mL^−1^ was obtained using DPV. Considering BSA, α-fetoprotein (AFP), and platelet-derived growth factor (PDGF-BB) as possible interfering proteins, their DPV current intensities showed no significant differences in a blank test. Reproducibility was acceptable in terms of RSD (4.74%). Then, aptasensor was applied to spiked real samples of human serum with recoveries ranging from 98.8 to 104.1%.

The next two examples reported the application of COF nanomaterials for CB determination. Notably, a review regarding COF application for determining disease biomarkers, including CBs, can be found in the literature. There are generally reported applications of COFs to all types of sensors for CB analysis, from optics to fluorescence sensors, but not necessarily to the electrochemical ones [136].

Cancer antigen 125 (CA-125) or Mucin 16 (MUC16) is a protein and biomarker produced by ovarian cancer cells. CA 125 normal values found in healthy women are in the range 0–35 U mL^−1^. Values higher than 35 U mL^−1^ are linked to ovarian cancer occurrence and development. Therefore, a CA-125 test could be useful to follow the ovarian cancer evolution during and after its treatment [137].

A sandwich immunosensor based on COF-LZU1 and multilayer reduced graphene oxide frame (MrGOF) was developed for assaying CA 125 [137]. COF-LZU1 was casted on GCE and acted as a platform for immobilizing CA 125 first antibody, while MrGOF, functionalized with an amino group and decorated with silver nanoparticles (AgNPs), acted as a probe to label CA 125 s antibody. A scheme of the immunosensor assembling is reported in Figure 17.

Under optimized conditions, CA 125 was determined using DPV, and a linear concentration range from 0.001 to 40 U mL^−1^ and LOD of 0.00023 U mL^−1^ were found. CA19-9 antigen, CA72-4 antigen, horseradish peroxidase (HRP), AA, and BSA were analysed as possible interferences, evidencing that they did not affect CA 125 determination. Long-term stability was investigated: the DPV signal decreased by 19.11% after 7 days at 4 °C. The immunosensor was applied to spiked samples of human serum, obtaining recoveries ranging from 91.54% to 105.21%.

Prostate-specific antigen (PSA) is a well-known biomarker for prostate cancer (PCa) diagnosis. PSA normal values found in healthy men are in the range 0–4 ng mL^−1^. Values higher than 4 ng mL^−1^ are related to PCa occurrence and development. Therefore, the PSA method to determine PCa accurately is very important [138].

A peptide-PSA-antibody sandwich immunosensor was developed using polydopamine-coated boron-doped carbon nitride, including AuNPs, (Au@PDA@BCN) and covalent organic frameworks (COF) functionalized with AuPt bimetallic nanoparticles and manganese dioxide (MnO_2_) (AuPt@MnO_2_@COF) [131]. Boron-doped carbon nitride (BCN) is a heteroatom-doped 2D carbon material with low toxicity and high stability, while PDA is a conducting polymer. A Au@PDA@BCN nanocomposite was deposited on a GCE and acted as a sensing platform to immobilize PSA primary antibodies. On the other hand, AuPt@MnO_2_@COF served as an electrocatalyst and signal-amplifier. PSA affinity peptide was immobilized on it to form a Pep/MB/AuPt@MnO_2_@COF nanocomposite, including MB as a redox indicator. PSA was determined using DPV, with a linear response in the range of 0.00005–10 ng mL^−1^ with LOD of 16.7 fg mL^−1^. AA, BSA, lysine, ovalbumin, glucose, lysozyme, sucrose, and lipase were considered as interferences because they are present together with PSA in the same complex matrices or structural analogues. The DPV response of a mixture of PSA and all the interfering molecules was similar to that produced by PSA. Repeatability was satisfactory in terms of RSD (1.6%); DPV response decreased by 5% with respect to its initial value after 14 days at 4 °C. Spiked serum samples were analysed with recoveries ranging from 98.9% to 100.2%.

An aptasensor based on a two-dimensional porphyrin-based covalent organic framework (p-COF) was developed, immobilizing epidermal growth factor receptor (EGFR)-targeting aptamer strands for determining trace EGFR and living Michigan Cancer Foundation-7 (MCF-7) cells [139].

Epidermal growth factor receptor (EGFR) is a transmembrane protein and its abnormal values (>75.3 mg·L^−1^) may be linked to different cancer typologies, such as lung, breast, prostate, bladder, colorectal, pancreatic, and ovarian.

MCF-7 is a human breast cancer cell line with estrogen, progesterone, and glucocorticoid receptors. It was isolated for the first time from the pleural effusion of a 69-year-old Caucasian metastatic breast cancer (adenocarcinoma) at the Michigan Cancer Foundation.

The two-dimensional p-COF structure can provide several binding sites for aptamers or biomolecules and its conjugated structure can improve electrochemical activity. In addition, the presence of large pore channels supports the aptamer immobilization inside the p-COF structure, thereby increasing the number of biomolecules adsorbed and further enhancing the sensor analytical performances. P-COF was deposited on AuE and EGFR was determined using EIS and DPV. The same linearity range was obtained via EIS and DPV (0.05–100 pg mL^−1^), while two different LOD values were found: 7.54 × 10^−3^ pg mL^−1^ (EIS) and 5.64 × 10^−3^ pg mL^−1^ (DPV). CEA, PSA, Mucin-1 (MUC1), human epidermal growth factor receptor 2 (HER2), vascular endothelial growth factor (VEGF), immunoglobulin G (IgG), platelet-derived growth factor-BB (PDGF-BB), and BSA, present together with EGFR in human serum, were tested as interfering biomolecules at a concentration 1000 times higher than that of EGFR. Their electrochemical response was negligible compared with that of EGFR. After 10 days at 4 °C in dry state, the electrochemical response remained almost stable. Reproducibility was acceptable in terms of RSD (2.29%). The aptasensor was applied to spiked human serum samples to determine EGFR with recoveries ranging from 96.2% to 103.2%. After testing the biocompatibility of the aptasensor with MCF-7 cells, the sensor was applied for determination in artificial samples MCF-7 cells, with LOD of 61 cells·mL^−1^ and a linear detection range of 5 × 10^2^–1 × 10^5^ cell·mL^−1^.

As final considerations, LOD values achieved generally pg mL^−1^ and the corresponding linearity ranges seem to be wide, considering the application field.

There is not a preferred sensor format and all the (bio)sensors have been applied to real samples. Comparison and validation with standard methods are, however, missing.

Analytical performance and format of the reported electrochemical biosensors for CBs determination are summarized in Table 5.

### 3.6. Antibiotics

Antibiotics are drugs used for treating bacterial diseases in animals and plants because they can destroy bacterial cells by either preventing cell reproduction or modifying necessary cellular function or processes within the cell [140]. Antibiotics are non-biodegradable compounds, and they can induce endocrine disorders, anaemia, mutagenicity, etc., if they remain in the human body. Consequently, it is mandatory to have effective, responsive, and fast antibiotic trace detection methods.

A recent review reported some examples of electrochemical sensors based on MOFs for the determination of antibiotics [140], describing MOFs’ synthetic methods and the different typologies of the included sensors. The most recent examples of electrochemical sensors for antibiotics have been herein reviewed, also considering other 2D nanomaterials.

Enrofloxacin (ENR) is a fluoroquinolone antibiotic used to fight bacterial diseases in livestock and aquaculture. Ampicillin (AMP) is a broad-spectrum antibiotic classified as β-lactam, widely used because of its ability to kill Gram-positive and -negative bacteria by destroying the cell wall.

Electrochemical aptasensors were developed for ENR and AMP analysis using AuE-modified with COF synthesized from the condensation-polymerization of 1,3,6,8-tetrakis(4-formylphenyl)pyrene and melamine through imine bonds (Py-M-COF) [141]. Py-M-COF presented an extensive π-conjugation framework, a large specific surface area, a nanosheet-like structure, different functional groups, and good conductivity, so a stable and firm aptamers immobilization was performed via π-π stacking and electrostatic interactions. ENR was determined using EIS and a linear concentration range of 0.01 pg mL^−1^–2 ng mL^−1^ with LOD of 6.07 fg mL^−1^ was found. Similarly, AMP was quantified with a linearity range of 0.001–1000 pg mL^−1^ and LOD of 0.04 fg mL^−1^. Tetracycline, kanamycin (Kana), tobramycin (TOB), streptomycin, and oxytetracycline (OTC) were tested as possible interfering antibiotics and the electrochemical response of all those vs. both ENR and AMP was negligible. The reproducibility of the two aptasensors was acceptable considering the RSD values, 1.25% and 1.44% for ENR and AMP, respectively. Long-term stability was investigated: the EIS response remained almost stable after 14 days at 4 °C. ENR and AMP were assayed in spiked samples of human serum and recoveries were in the range of 101.0–112.4% for ENR and 99.5–103.0% for AMP.

Tetracycline (TC) is a broad-spectrum antibiotic for the treatment of bacterial infections in humans and animals and is used as animal feed-additive to prevent animal infections because of its low cost and high antibacterial activity. Notably, antibiotic residues in food can cause health problems and induce bacterial resistance to TC in humans and animals.

A molecularly imprinted tetracycline electrochemiluminescence (ECL) sensor was assembled based on Zr-coordinated amide porphyrin-based 2D COF (Zr-amide-Por-based 2D COF) [142]. Zr-amide-Por-based 2D COF was dropped onto GCE and then the TC-molecularly imprinted electrochemiluminescence sensor (TC-MIECS) was assembled to perform the electropolymerization of o-phenylendiamine (o-PD), acting as a functional monomer, and using TC as a molecular template. After removing the template molecule, imprinted cavities serving as TC recognition elements were realized. Under optimized experimental conditions, TC-MIECS showed a linear relationship with tetracycline in the concentration range 5–60 pM, with LOD of 2.3 pM. The sensor evidenced an acceptable repeatability in terms of RSD (6.7%). Operative stability was considered satisfactory because, after 12 measurements, the RSD value of the ECL intensity variation was 0.94%. Chloramphenicol (CAP), OTC, AMP, and penicillin were investigated as interfering antibiotics and the ECL intensity response of all the interferences vs. that of TC was negligible because of the high specificity of the molecularly imprinted cavities. TC-MIECS were applied to spiked real samples of milk and recoveries ranging from 94.0% to 103.5% were obtained.

Chloramphenicol (CAP) is a broad-spectrum antibiotic used against the main species of Gram-positive and -negative bacteria, as well as other groups of microorganisms such as *Salmonella*. Since CAP can trigger several collateral effects, such as anaemia or mutagenicity in humans, its use is limited and controlled or even forbidden in animal husbandry for food production in many countries [143].

An electrochemical aptasensor based on a nanocomposite including Zirconium-porphyrin MOF (PCN-222) and graphene oxide (PCN-222/GO) was prepared for the determination of CAP [144]. The high conductivity of GO, together with MOF mesoporous channels and metal sites, facilitated a stable aptamer immobilization owing to a π−π stacking interaction and the connection between the aptamer phosphate group and Zr(IV) sites of PCN-222. PCN-222/GO was dropped on AuE and then the aptamer was adsorbed on the nanocomposite. CAP was determined using EIS and a linear concentration range from 0.01 to 50 ng mL^−1^ and LOD of 7.04 pg mL^−1^ were found. OTC, TC, kanamycin sulphate (KS), metronidazole (MDZ), and nitrofurantoin (NFT) were used as interfering compounds, with the EIS response of all of them being insignificant compared with that of CAP. Reproducibility was acceptable in terms of RSD (6.29%), and long-term stability was defined as satisfactory without indicating storage conditions. The aptasensor was applied to spiked real samples of milk, human serum, and urine with recoveries ranging from 94.6 to 107.2%.

Norfloxacin (NF) presents a broad-spectrum antibacterial activity and, for this reason, it has been used for the treatment of human and animal diseases. However, its excessive use can trigger antibiotic resistance, collateral effects, and toxicity. Residual amounts or traces of NF can remain in foods and tissues because of its uncompleted metabolism. Therefore, the production and use of NF have been limited and controlled.

β-CD porous polymers (P-CDPs) functionalized with COF (PCDPs/COFs) and combined with Pd^2+^ via electrostatic interaction (Pd^2+^@P-CDPs/COFs) were synthesized and casted onto GCE to assemble a non-enzymatic electrochemical sensor for the assay of NF [145], as shown in Figure 18.

COF was prepared starting from 1,3,5-tris(4-aminophenyl)benzene (TAPB) and 2,5-dimethoxyterephaldehyde (DMTP) via the simple solution infiltration method. β-CD entrapped NF through a host–guest interaction, while COF interacted electrostatically via -NH_2_ functionalities with the negatively charged functional group of NF; consequently, the adsorption of NF onto the modified electrode improved. Moreover, Pd^2+^ increased the nanocomposite catalytic performances and acted as an electrochemical signal amplifier. After experimental conditions optimization, the NF was determined using DPV, and two linear concentrations in the ranges of 0.08–7.0 μM and 7.0–100.0 μM with LOD of 0.031 μM were evidenced. Precision was verified and the results in terms of RSD were in the range of 2.91–3.46% for intra-day precision and 3.13–3.47% for inter-day precision. Concerning accuracy, the results obtained for intra-day and inter-day accuracy were < 3.00%. Glucose, AA, UA, DA, and Levofloxacin (LF) were evaluated as interference molecules and none of them affected the electrochemical response of the NF. Repeatability and reproducibility were considered acceptable in terms of RSD, 2.9% and 3.1%, respectively. The sensor was applied to determine NF in the spiked samples of Norfloxacin Eye-drops. The results were comparable to those coming from a standard method using HPLC, with recoveries ranging from 97.8% to 101.7%.

The next three examples introduce electrochemical sensors for the determination of three antibiotics belonging to the sulphonamide family with a wide spectrum of antimicrobial activities against protozoa and bacteria. The first one is sulfathiazole (STZ), well-known as being low cost and a broad-spectrum antibiotic, widely employed for the treatment of animal and human infections. On the other hand, its accumulation in humans and livestock causes serious effects on hematopoietic systems by blocking the dihydrofolic acid synthesis [146], so its use is limited and controlled especially to guarantee food safety.

A GCE-modified electrochemical MIP sensor for the determination of STZ was designed to exploit CuS microflowers as redox probe polypyrrole as MIP, imprinted with STZ, and AuNPs incorporated in a COF structure (Au@COF) for conductivity [147]. The scheme for MIP-sensor assembling is shown in Figure 19.

STZ was quantified using DPV, and a decreasing electrochemical response was observed for increasing STZ concentration, owing to selective STZ adsorption on the MIP film. Consequently, the corresponding CuS electron transfer at the sensing interface was hampered. The DPV current decrease was directly proportional to the logarithm of STZ concentration over a concentration range from 0.001 to 100,000 nM with LOD of 0.0043 nM. Structural analogues of STZ, such as sulfadimidine, sulfacetamide, and sulfadiazine (SDZ), and glucose, glutamate, and AA were tested as possible interfering molecules, evidencing that they did not interfere with the STZ assay. Reproducibility was considered, with acceptable results in terms of RSD (4.5%); DPV response decreased by 15.4% after 30 days, without specifying storage conditions. Mutton and fodder spiked samples were used to investigate accuracy and the applicability to real samples, with recoveries in the range of 83.0–107.3%. The MIP sensor was applied to samples of chicken liver and pig liver and the results were comparable to those coming from HPLC as a reference method, indicating no significant difference in amounts determined by the two methods (*p* > 0.05).

Sulfamethoxazole as STZ can cause haematological problems, as well as hypersensitivity and gastrointestinal diseases. Moreover, sulfamethoxazole traces in food with animal origin can induce thyroid cancer and several other diseases [148].

An electrochemical sensor based on CPE modified with a Fe_3_O_4_/ZIF-67 nanocomposite and ionic liquid (IL) 1-Butyl-3-methylimidazolium hexafluorophosphate (Fe_3_O_4_/ZIF-67/ILCPE) was prepared for the determination of sulfamethoxazole [149]. ZIF-67 is a well-known MOF with peculiar catalytic properties and chemical and thermal stability, as described above, while Fe_3_O_4_ can improve conductivity and electron-transfer to and from the electrode surface. Under the optimized experimental conditions, the antibiotic was determined using DPV, with the current intensity directly proportional to the concentration over the range 0 0.01–520.0 μM with LOD of 5 nM. River and tap water and urine-spiked samples were used to investigate the sensor applicability to real samples, and recoveries in the range of 97.1–103.3% were found. The reproducibility, repeatability, and stability data of the sensor were not provided.

Sulfadiazine (SDZ) is an antibiotic used for the treatment of several infections in animals and humans. SDZ inhibits bacteria growth by interfering with folate metabolism, so its use is regulated and controlled especially for food safety [150].

An electrochemical sensor for the simultaneous determination of SDZ and AP was developed based on an MIP for the recognition of SDZ and AP, and a GO@COF nanocomposite for signal amplification [150]. The nanocomposite was first deposited on GCE; a polypyrrole MIP was then electropolymerized on the modified electrode. MIP synthesis was performed via pyrrole electropolymerization in the presence of the template molecules. Then, they were removed just after the electropolymerization ended. Under optimized experimental conditions, SDZ and AP were determined simultaneously through DPV. Linear concentration ranges of 0.5–200 μM for SDZ and 0.05–20 μM for AP, with LODs of 0.16 μM and 0.032 μM, respectively, were found. Molecules structurally similar to SDZ, such as sulfamerazine and sulfacetamide, and to AP, such as AA and p-nitrophenol, were assumed as possible interferences. The electrochemical response of the modified GCE towards SDZ and AP were higher than those of the respective structural analogues. Reproducibility was satisfactory in terms of RSD: 5.5% (SDZ) and 6.7% (AP). Concerning repeatability, RSD values of 2.7% for SDZ and 5.8% for AP were found. The DPV response decreased by 17.6% for SDZ and 14.6% for AP, after 30 days, without specifying the storage conditions. The sensor was applied to spiked samples of fodder and beef, extracted with an organic solvent, for investigating its accuracy. Recoveries ranging from 82.0% to 108.0% were obtained. The sensor was applied to real samples of pork and chicken, with data comparable to those coming from the HPLC analysis.

Tobramycin (TOB) is an effective antibiotic commonly used in the treatment of various systemic and ocular infections. As an aminoglycoside antibiotic, tobramycin is active against aerobic gram-negative bacteria. Like other aminoglycosides, several side effects for tobramycin can be evidenced such as ototoxicity, nephrotoxicity, and neuromuscular toxicity.

An impedimetric aptasensor based on a Zirconium-porphyrin MOF PCN-222 (Fe) nanosheet (PCN-222(Fe)-NS) was assembled to analyse TOB [151]. PCN-222(Fe)-NS was prepared, starting from [5,10,15,20-tetrakis(4-carboxyphenyl) porphyrinato]-Fe(III) chloride and Zr clusters, and supported the aptamer immobilization via π–π interaction and coordination bond. Under optimized conditions, a linear range of 1.1 × 10^−4^–10.7 nM and LOD of 1.3 × 10^−4^ nM were determined. Considering the aptasensor reproducibility, RSD values smaller than 4.92% were obtained. Penicillin G (PCL), oxytetracycline (OTC), doxycycline (DOX), streptomycin (STR), and kanamycin (KAN) were investigated as possible interfering antibiotics, without affecting the impedimetric response of TOB. The electrochemical signal was almost stable after 7 days of storage at −20 °C. The aptasensor was applied to spiked real samples of milk and recoveries ranging from 97.5% to 104.8% were found.

A covalent organic framework nanosheet (COF NS) synthesized from 2,4,6-triformylphloroglucinol (Tp) and 5,5′ -diamino-2,2′ -bipyridine (Bpy) (Tp-Bpy COF NS) was used as functional material to immobilize the aptamer for assembling an impedimetric aptasensor to determine TOB [152]. Tp-Bpy COF NS showed a high surface area and stability and different functional sites to support the aptamer immobilization. TOB was assayed by EIS and a linear concentration range of 2.1 × 10^−4^–10.7 nM and LOD of 6.57 fM were obtained. Reproducibility was investigated by using five Tp-Bpy COF NS aptasensors, with RSD values lower than 4.17%. Tetracycline (TET), ampicillin (AMP), oxytetracycline (OTC), kanamycin (KAN), penicillin G (PCL), doxycycline (DOX), and ornidazole (ODZ) were tested as possible interfering antibiotics, which did not affect the EIS response of the TOB. The electrochemical signal was almost stable after 10 days storage at −20 °C. The aptasensor was applied to spiked real samples of milk and river water, with recoveries in the range of 95.4–105.2% in milk and of 96.1–104.4% in river water.

Considering all the sensors described, the LOD values achieved concentration levels pM or nM, and the corresponding linearity ranges seem to be wide enough considering the application field. The preferred (bio)sensor format seems to be aptasensor. Validation with standard methods was performed for three examples only [145,147,150].

The analytical performance and format of the reported electrochemical biosensors for antibiotics determination are summarized in Table 6.

## 4. Conclusions

In this section, some considerations and comments regarding the role of 2D nanomaterials and the different types of electrodes and biosensors are summarized.

In addition, some comments concerning biosensors’ analytical performances, such as linearity range, detection limits, selectivity, and validation with standard methods of analysis, along with the possibility of determining several analytes at the same time or the applicability in real matrices, are introduced. Finally, critical issues, challenges, and future perspectives on the electrochemical (bio)sensing approach for biomedical applications involving 2D nanomaterials are outlined.

In most cases, the most used nanomaterial was found to be MXenes because of their high electrochemical activity and conductivity, as well as large surface area and well-established synthetic procedures [16,17,18,19,20]. In addition, MXenes can be easily functionalized and, consequently, their performance in the electrochemical sensing area relies on the type and number of these functional groups. Finally, MXenes nanohybrids including different nanomaterials or polymers can be easily synthesized, and the sensing properties of the nanocomposite can be enhanced with respect to those of the starting material.

MOFs are also largely employed in (bio)sensors for biomedical applications, probably because it is possible to tune their structure and functionality [30,31], and include in the MOF’s structure different conducting nanomaterials such as metal nanoparticles can improve the poor conductivity and sensing properties of the starting MOFs.

COFs and TMDs are much less likely to be used because COFs’ synthesis methods still need to be optimized, while synthetic approaches for TMDs require drastic conditions in terms of temperature and pressure, which do not facilitate their use to design (bio)sensors for biomedical applications.

The integration of nanomaterials in the development of electrochemical (bio)sensors is crucial for improving their analytical performances.

Nanocomposites and/or nanohybrids represent the preferred sensing interface, including different nanomaterials or synthetic polymers, and sometimes the resulting nanostructure with tailored architecture can be very complex.

Considering the electrode typologies reported in this review (see Table 1, Table 2, Table 3, Table 4, Table 5 and Table 6), different types of electrodes are employed for the determination of analytes of clinical and biomedical interest, from more conventional bulk electrodes such as GCE, AuE, and ITOE, to IDEA (interdigitated electrode array), IGE (interdigitated gold electrode), CCE (carbon cloth electrode), and SPEs, among others. GCE was the preferred option probably because of its well-known chemical–physical properties [3].

Gold electrode (AuE) and SPEs were employed in several examples because Au is a biocompatible, stable, easy to functionalise, and a conducting material, and SPEs represent low-cost sensing platforms, able to move the transition from conventional laboratory equipment to portable devices.

Regarding different types of (bio)sensors, it is difficult to make general considerations, because examples with different types of analytes have been examined. In the case of glucose, neurotransmitters and hormones as target molecules, chemosensors, are prevailing, the key recognition element being generally a nanocomposite including 2D nanomaterials [see Table 1, Table 2 and Table 3]. In the case of glucose, only a few examples are represented by classical enzymatic biosensors based on glucose oxidase [see Table 1].

On the other hand, when pathogens, CBs, or antibiotics are the target analyte, immunosensor or aptasensor are the prevalent type [see Table 4, Table 5 and Table 6].

The number of examples using molecularly imprinted polymer as a key recognition element was limited [see Table 3, Table 4 and Table 6], even if it could represent a promising synthetic receptor to be used instead of more conventional ones.

Analytical performances in terms of linearity range and/or LOD seem to be very promising: in fact, several examples involved nanomolar concentrations with the LOD at a femtomolar level, depending on the target. Regarding selectivity, this issue is generally addressed, but the criterion of choice of interfering compounds is often not clear, even for the same target. It would be useful to indicate this criterion: interferences might have been chosen because they have similar structure or function or because they are present in the same complex matrices to be analysed where the target is also present, just to suggest some examples.

The reproducibility, repeatability, and stability of the sensors have been generally investigated. However, it is difficult to compare data from different sensors, even when regarding the same target, because comparable experimental procedures are not used. In particular, referring to long-term stability, storage conditions in terms of temperature and wet or dry storage are not always indicated.

Validation with standard methods is rarely performed, even if this step is mandatory in order to have a clear indication of the sensors’ analytical performances in comparison with those of more conventional approaches.

Moreover, it should be stressed once again that electrochemical (bio)sensors are generally not commercially available except for the glucometer [see Table 1]. Different problems and challenges are involved, such as complex and expensive material synthesis, samples preparation and stability, unforeseen interfering molecules, collateral chemical reactions in real matrices, or the fouling and biofouling of the electrode surface.

Finally, coming back to 2D nanomaterials, greater attention should be paid to the relationship among structure and material properties, and which 2D nanomaterial properties really affect the (bio)sensor performances.

All these challenges should be addressed and solved in view of the electrochemical (bio)sensor introduction into the market, because the industrial interest in their commercialization is connected to market demand considering all the costs associated with new technologies. Consequently, an interdisciplinary approach and the close collaborations of analytical chemists, materials scientists, biochemists, engineers, and physicians can effectively support solving these problems.

## Figures and Tables

**Figure 1 molecules-29-00172-f001:**
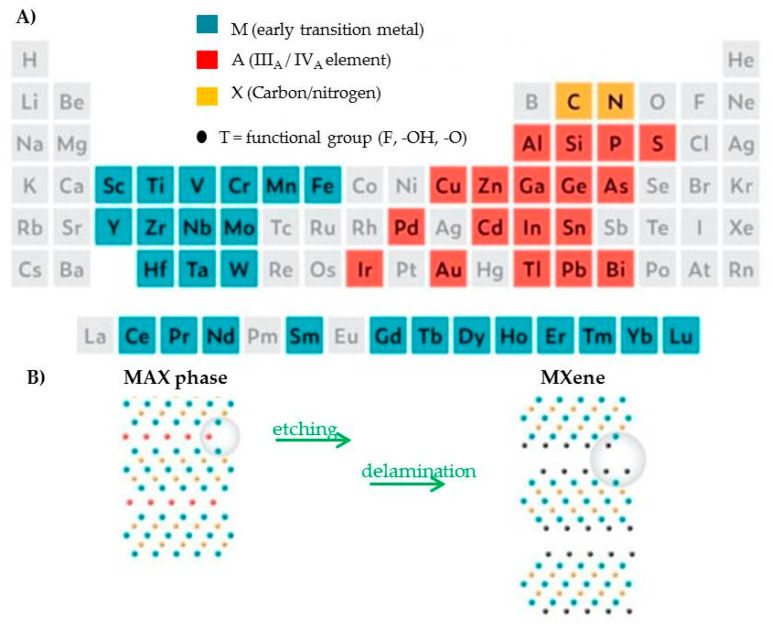
MXenes: (**A**) chemical composition starting from MAX and (**B**) schematics of the synthetic path.

**Figure 2 molecules-29-00172-f002:**
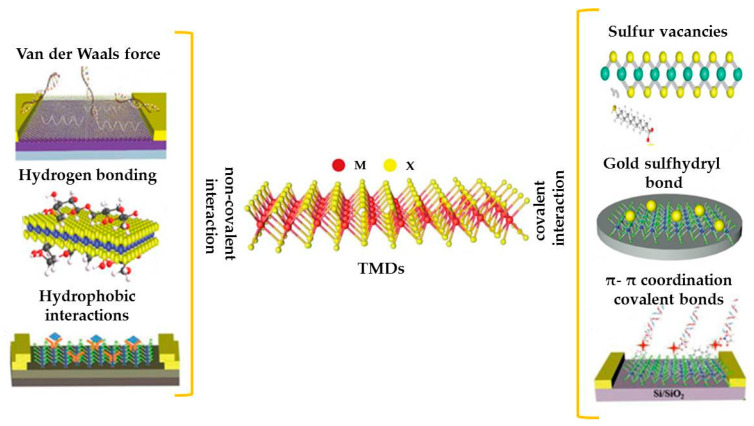
TMDs: structure (center), M = transition metal and X = chalcogen; structures and surface modification: non-covalent (**left**) and covalent (**right**) interactions.

**Figure 3 molecules-29-00172-f003:**
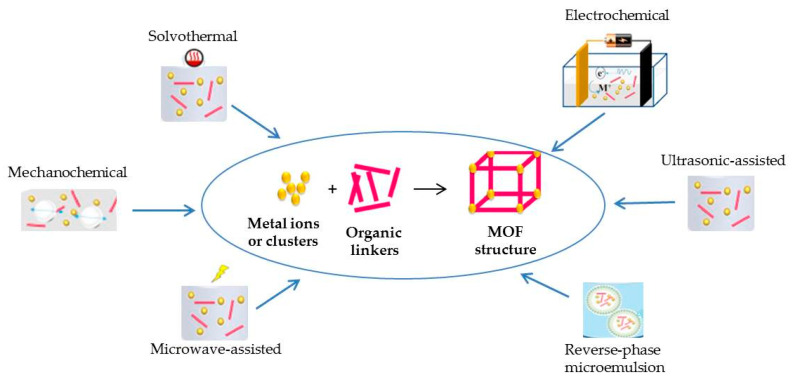
MOFs: synthetic strategies.

**Figure 4 molecules-29-00172-f004:**
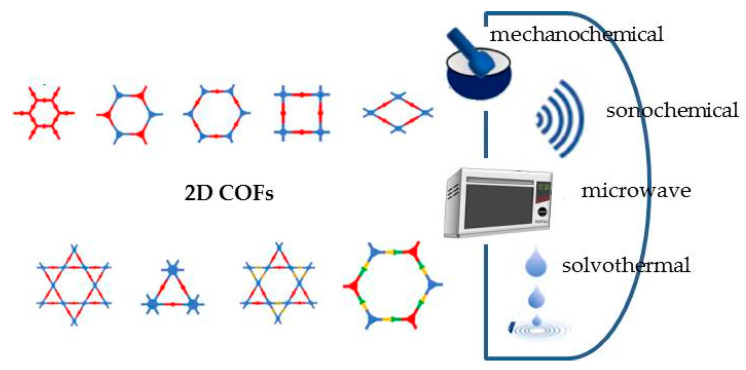
Two-dimensional COFs: structures and synthetic strategies.

**Figure 5 molecules-29-00172-f005:**
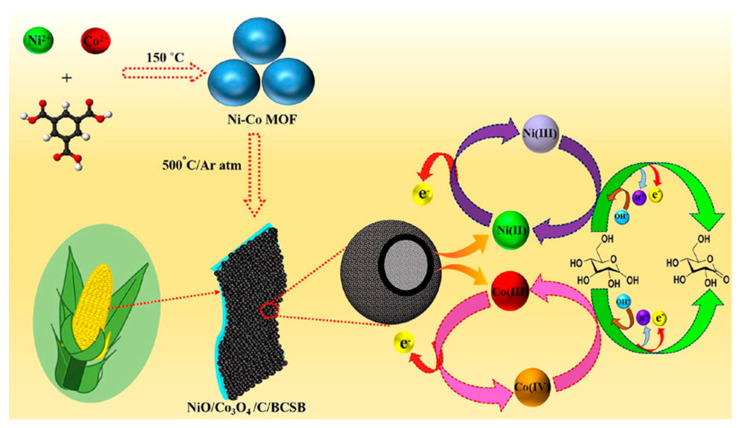
Schematic representation of Ni-Co MOF synthesis, a non-enzymatic glucose-sensor-assembling and glucose-detection mechanism, reprinted with permission by [57] Copyright 2022, Elsevier. The arrows indicate the well-known glucose detection mechanism.

**Figure 6 molecules-29-00172-f006:**
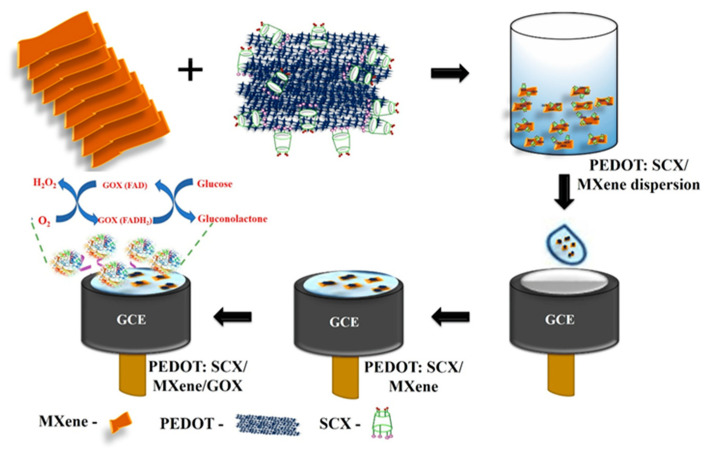
Scheme of GCE modification for assembling glucose biosensor reprinted from [64]. The arrows indicate the several steps of the glucose sensor assembling.

**Figure 7 molecules-29-00172-f007:**
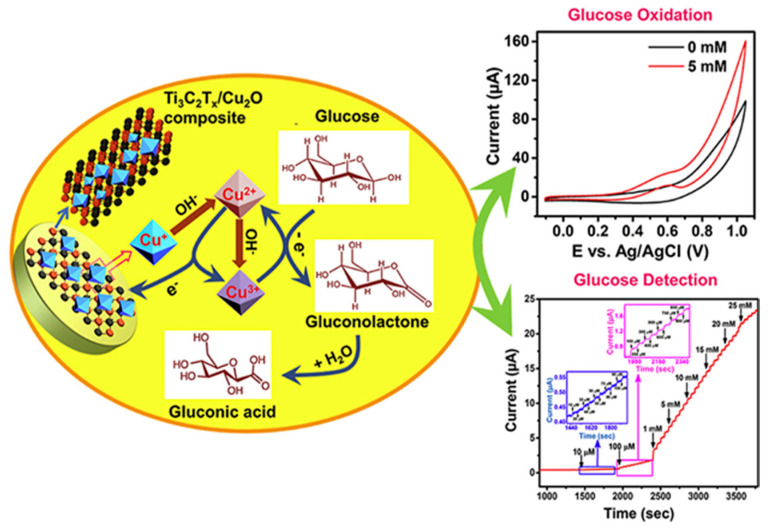
Schematic representation of electrode modification and glucose sensing mechanism, reprinted with permission from [70] Copyright 2022 Elsevier. The arrows indicate the cyclic volyammetry of the glucose oxidation and the glucose detection by chrono amperometry.

**Figure 8 molecules-29-00172-f008:**
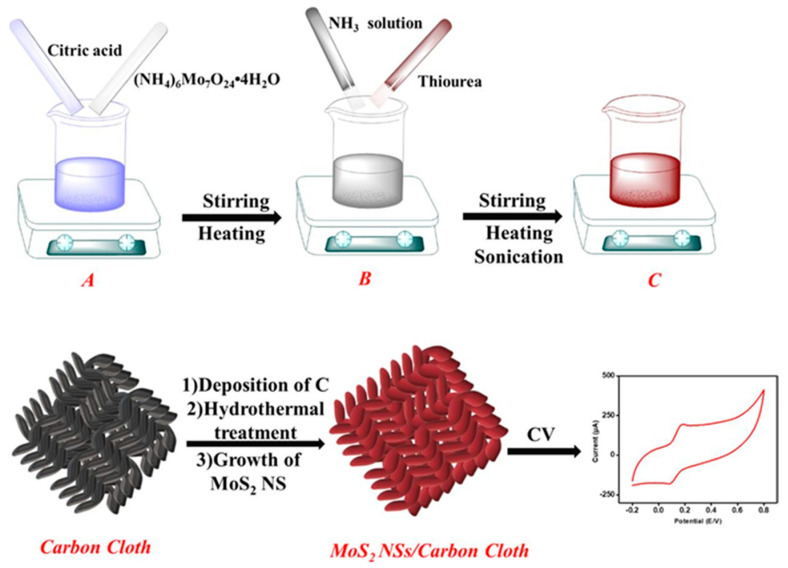
Scheme of TMD synthesis, CC electrode modification and electrochemical characterization, reprinted with permission from [77] Copyright 2022 Elsevier.

**Figure 9 molecules-29-00172-f009:**
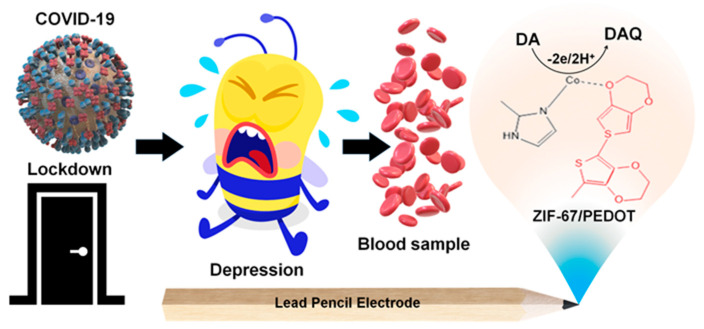
Scheme of the sensing mechanism involved in DA assay at ZIF-67/PEDOT-modified LPE, reprinted with permission from [80] Copyright 2022 Elsevier.

**Figure 10 molecules-29-00172-f010:**
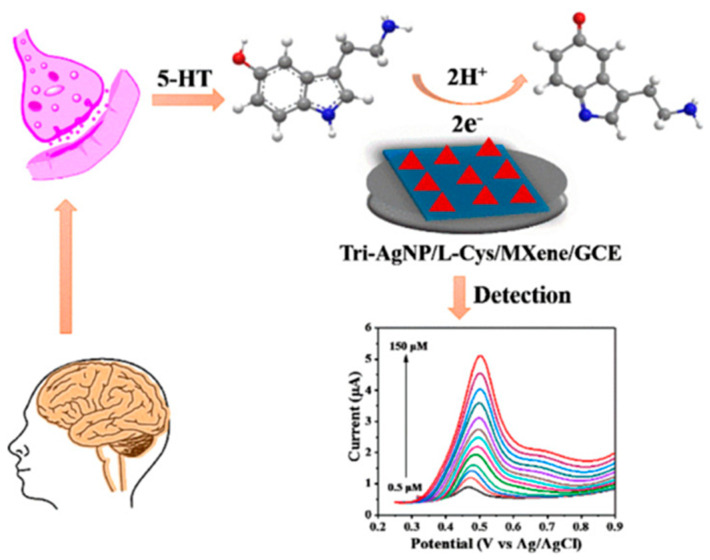
Sensing mechanism of 5-HT determination at MXene-modified GCE, reprinted with permission from [92] Copyright 2021 American Chemical Society. The colored lines correspond to the different electrochemical signals using different concentration of 5-HT.

**Figure 11 molecules-29-00172-f011:**
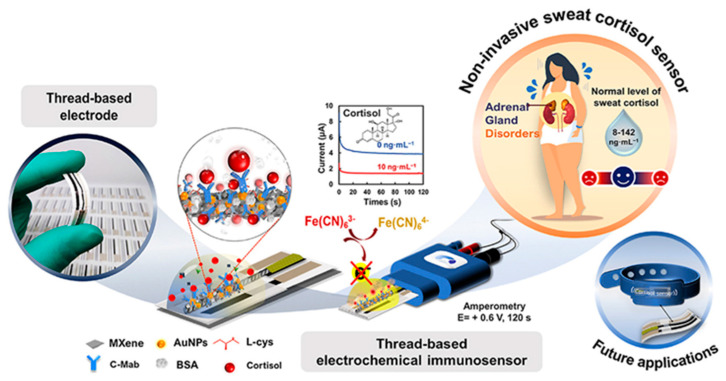
Schematic representation of L-cys/AuNPs/MXene-modified conductive thread electrode assembling and sensing mechanism of cortisol determination, reprinted with permission from [99] Copyright 2022 Elsevier.

**Figure 12 molecules-29-00172-f012:**
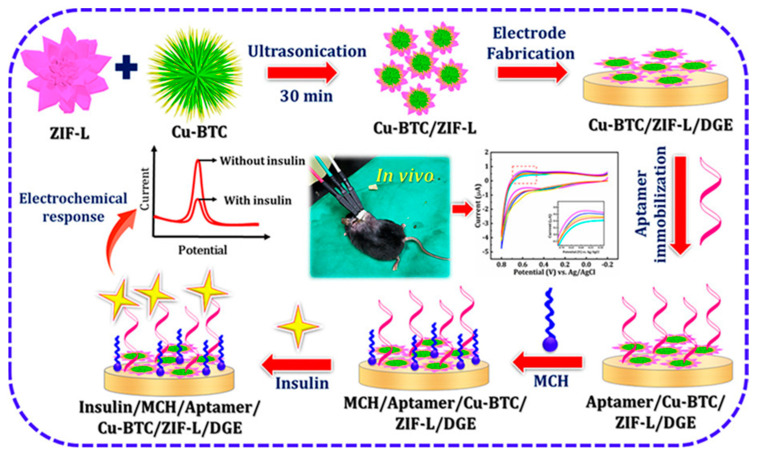
Assembling of Cu-BTC/ZIF-L composite-modified DGE and sensing mechanism of insulin assay, reprinted with permission from [105] Copyright 2022 American Chemical Society.

**Figure 13 molecules-29-00172-f013:**
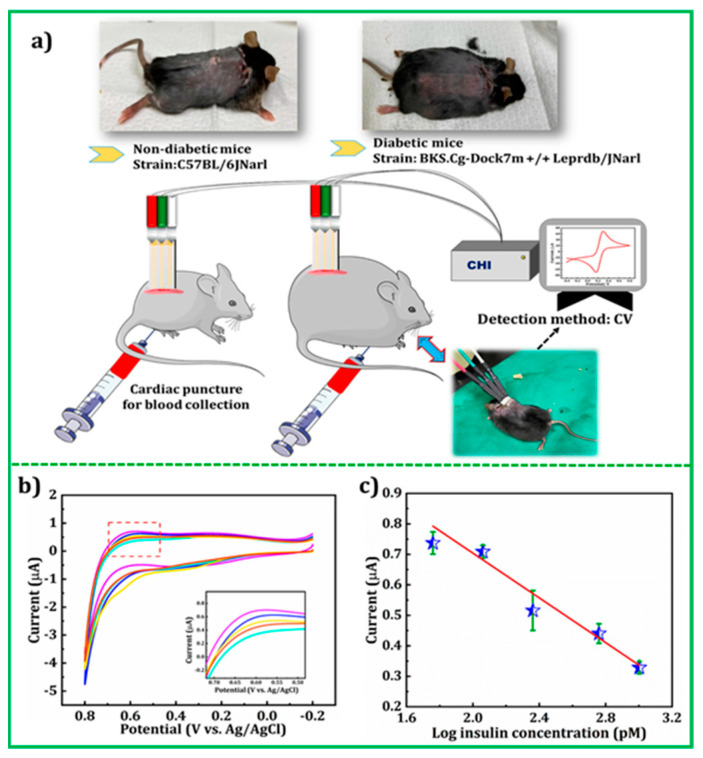
(**a**) Schematic of real-time in vivo insulin monitoring using aptamer/Cu-BTC/ZIF-L/DGE. (**b**) CV curves of insulin concentration in serum samples determined by insulin aptasensor. (**c**) Calibration plot of anodic peak current vs. insulin concentration logarithm, reprinted with permission from [105] Copyright 2022 American Chemical Society.

**Figure 14 molecules-29-00172-f014:**
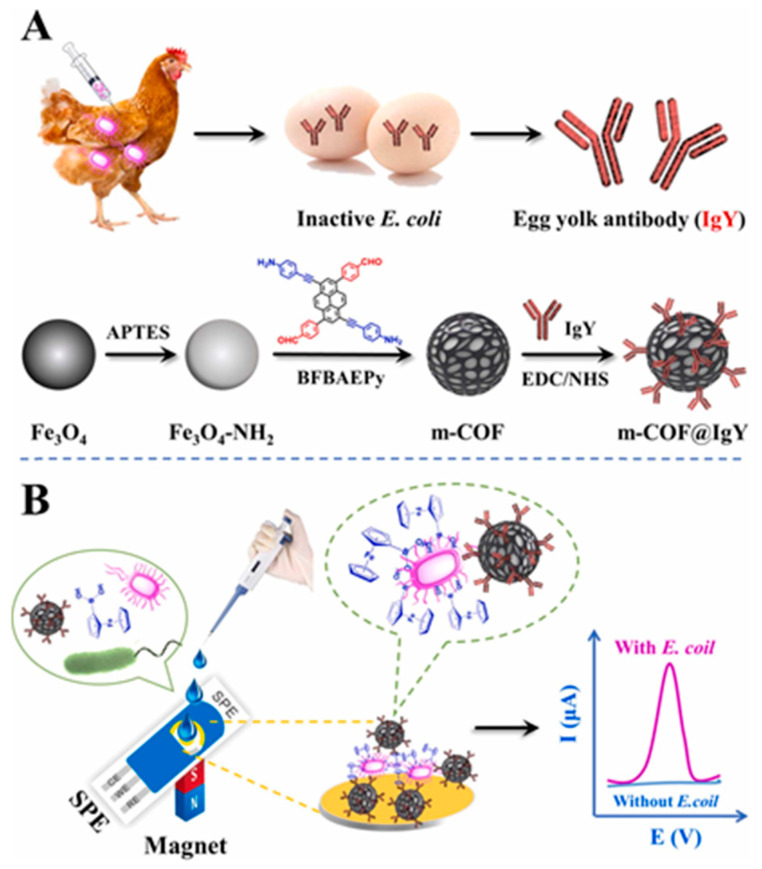
(**A**) Scheme for the synthesis of m-COF@IgY and IgY for *E. coli* and (**B**) sensing mechanism of the immunosensor for *E. coli* based on m-COF@IgY and ferrocene boric acid (FBA) using magnetic control screen-printed electrode (SPE) as detection platform, reprinted with permission from [108], Copyright 2022 Elsevier.

**Figure 15 molecules-29-00172-f015:**
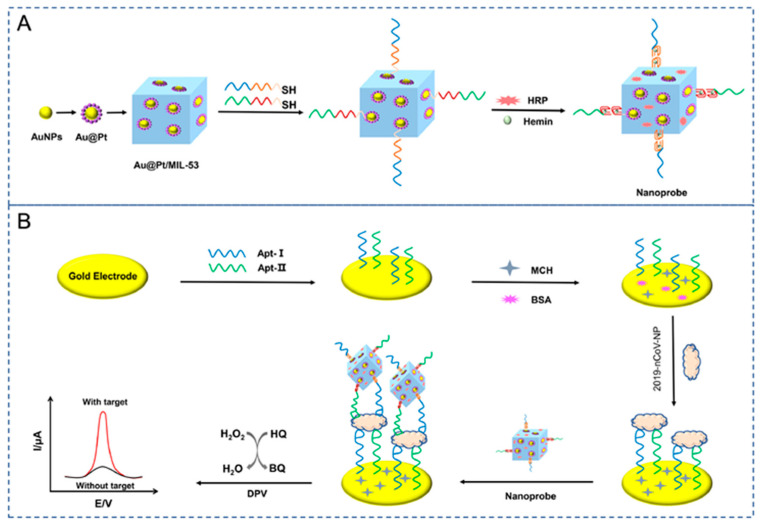
(**A**) Scheme of nanoprobe assembling and (**B**) sensing mechanism of the aptasensor for the detection of SARS-CoV-2 NP, reprinted with permission from [123], Copyright 2021 Elsevier.

**Figure 16 molecules-29-00172-f016:**
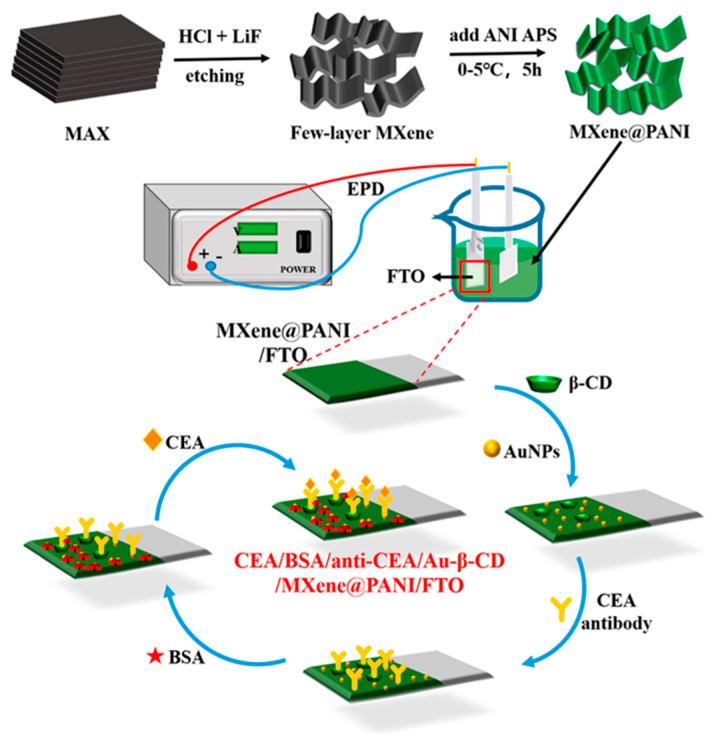
Scheme of the electrochemical immunosensor assembling steps for label-free detection of CEA, reprinted from [134].

**Figure 17 molecules-29-00172-f017:**
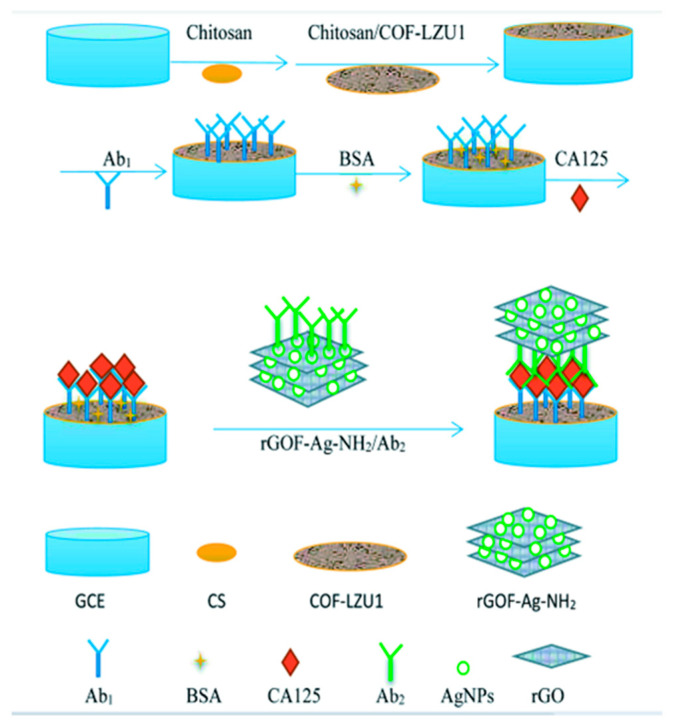
Scheme of the electrochemical sandwich immunosensor assembling for the determination of CA 125, reprinted with permission from [137] Copyright 2022 Elsevier.

**Figure 18 molecules-29-00172-f018:**
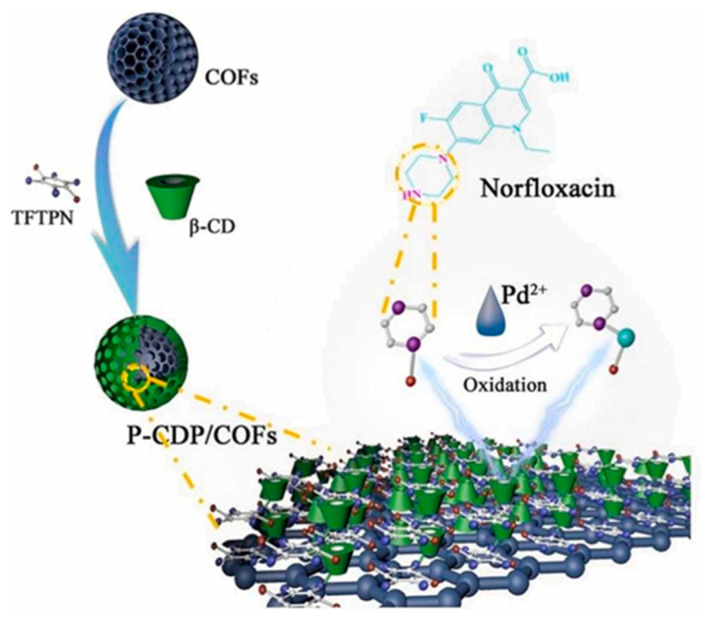
Scheme of non-enzymatic electrochemical sensor assembling for the determination of NF, reprinted with permission from [145] Copyright 2022 Elsevier.

**Figure 19 molecules-29-00172-f019:**
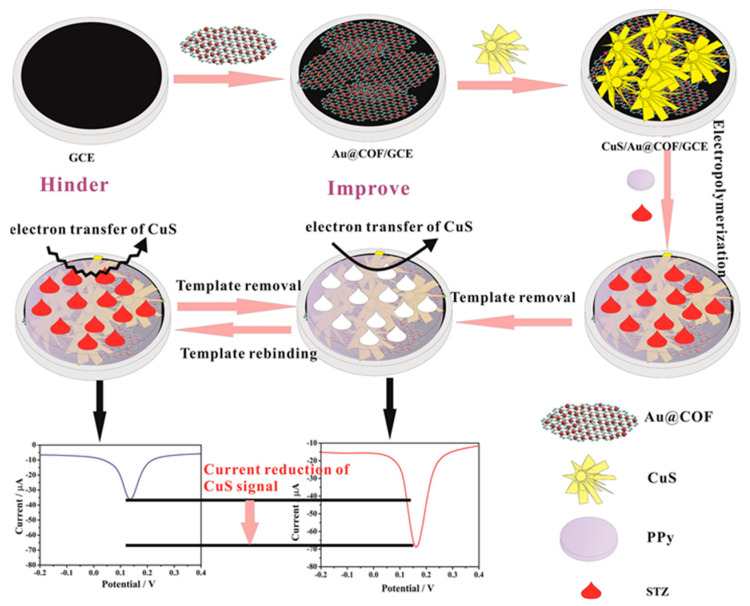
Scheme for MIP/CuS/Au@COF/GCE-sensor assembling and the STZ-detection strategy, reprinted with permission from [147] Copyright 2020 Elsevier.

**Table 1 molecules-29-00172-t001:** Analytical performances and format of electrochemical (bio)sensors for glucose determination.

Electrode	2D Nanomaterial	Format	Technique	Sample	Linearity (μM)	LOD (μM)	Recovery %	Reference Method	Ref.
GCE	ZIF-67	sensor based on Ag@ZIF-67 nanocomposite	A	-	2–1000	0.66	-	-	[50]
GCE	ZIF-67	sensor based on Ag@TiO_2_@ZIF-67 nanocomposite	A	-	48–1000	0.99	-	-	[51]
rGO/PU fiber	Ni–Co MOF	sensor based on Ni–Co MOF/Ag nanocomposite	A	Human sweat	10–660	3.28	-	Glucometer blood test	[52]
SPCPE	NC-ZIF	biosensor based on GOX/Hemin@NC-ZIF	A	Human sweat	50–600	2	-	Glucometer blood test	[53]
CC	ZIF-67	sensor based on ZIF-67@GO/NiCo_2_O_4_	A	-	0.3–157.4	0.16	-	-	[54]
NF	Cu_1_Co_2_-MOF	sensor based on Cu_1_Co_2_-MOF	CA	-	50–500	23	-	-	[55]
GCE	Ni-MOF	sensor based on Ni-MOF@Ni-HHTP-5 NSs core@shell structures	A	-	500–2,665,500	48.5	-	-	[56]
BCSB	Ni-Co MOF as sacrificial template	sensor based on NiO/Co_3_O_4_/C nanocomposite	A	Human blood serum	0.2–10,000	0.045	98.3–102.4	-	[57]
CFE	Ni-MOF	sensor based on Ni-MOF/rGO nanocomposite	A	Orange juice	6–2090	0.6	-	-	[59]
GCE	N-doped-Co- MOF	sensor based on N-Co-MOF@PDA-Ag nanocomposite	A	Human serum	1–2000	0.5	96–110	-	[60]
AuE	Ni-Co-MOF	sensor based on Ni–Co MOF/Au/PDMS nanocomposite	A	Human sweat	20–790	4.25	-	Glucometer blood test	[61]
SPCE	MXene (Ti_3_C_2_Tx)	biosensor based on ZnO TPs/MXene/GOX	CA	Human sweat	50–700	17	-	Glucometer blood test	[62]
Cr/AuE	MXene (Ti_3_C_2_Tx)	biosensor based on MXene/GOX	A	-	100–10,000	12.1	-	-	[63]
GCE	MXene (Ti_3_C_2_Tx)	biosensor based on PEDOT:SCX/MXene/GOX	A	Fruit juice	500–8000	22.5	96–99	-	[64]
AuE	MXene (Ti_3_C_2_Tx)	biosensor based on GOX/PtNPs/NPC/MXene	CA	Human sweat	3–21,000	7	-	Glucometer blood test	[65]
GCE	MXene	sensor based on MXene/CHI/Cu_2_O	CV	Human serum	52.4–2000	52.4	98.04–102.94	-	[66]
GCE	MXene (Ti_3_C_2_Tx) ZIF-67	sensor based on Ti_3_C_2_Tx/ZIF-67 nanocomposite	A	-	5–7500	3.81	-	-	[67]
GCE	Ti_2_C-MXene	sensor based on Ti_2_C-TiO_2_-MXene nanocomposite	DPV	Human serum	0.1–200	0.12	99.80–100.23	Glucometer blood test	[68]
CFE	MXene	sensor based on Co_3_O_4_/MXene nanocomposite	A	Human blood serum, urine	0.05–7440	0.010	97.80–101.60	Glucometer blood test	[69]
GCE	MXene (Ti_3_C_2_Tx)	sensor based on MXene-Cu_2_O nanocomposite	CA	Human serum	10–30,000	2.83	-	Glucometer blood test	[70]
AuE	c-MOF (Cu_3_(HHTP)_2_)	sensor based on c-MOF	A	-	0.2–7 mM	10	-	-	[71]

Abbreviations: A: amperometry; BCSB: biodegradable corn starch bag; CA: chronoamperometry; CC: carbon cloth; CFE: carbon fiber electrode; CHI: chitosan; c-MOF: conjugated MOF; COF: covalent organic framework; DPV: differential pulse voltammetry; GCE: glassy carbon electrode; GO: graphene oxide; GOX: glucose oxidase; HHTP: 2,3,6,7,10,11-hexahydroxytriphenylene; MOF: metal–organic framework; NF: nickel foam; NPC: nanoporous carbon; NSs: nanosheets; PDMS: polydimethylsiloxane; PEDOT: poly(3,4-ethylenedioxythiophene; PtNps: platinum nanoparticles; rGO: reduced graphene oxide; SCX: 4-sulfocalix [4]arene; ZnO TPs: ZnO tetrapods.

**Table 2 molecules-29-00172-t002:** Analytical performances and format of electrochemical sensors for neurotransmitters determination.

Electrode	2D Nanomaterial	Format	Technique	Sample	Linearity	LOD	Recovery (%)	Ref.
GCE	MXene (Ti_3_C_2_T_x_)	sensor based on Ti_3_C_2_T_x_/PtNPs	A	-	Up to 750 μM	10 nM	-	[74]
PGS	TMD (MoS_2_)	sensor based on MoS_2_	DPV	DA	0.05–5 nM 5 nM–5 mM	50 pM	-	[75]
GCE	MXene (Ti-C-T_x_)	sensor based on Ti-C-T_x_	DPV	DA,AA,UA	DA: 0.5–4 μM AA: 0.5–50 μM UA: 0.1–1.5 μM	DA: 0.06 μM AA: 4.6 μM UA: 0.075 μM	-	[76]
CC	TMD (MoS_2_)	sensor based on MoS_2_ nanosheets	CV	DA	250–4000 μM	0.3 μM	-	[77]
GCE	TS-COF	sensor based on AuNPs@TS-COF/rGO	DPV	DA,AA,UA in human urine	DA: 20–100 μM AA: 8–900 μM UA: 25–80 μM	DA: 0.03 μM AA: 4.30 μM UA: 0.07 μM	97–104.4	[78]
GCE	Fe-based MOF MIL-100(Fe)	sensor based on POM- MOF/PVP	DPV	DA,UA in human serum	DA: 1–247 μM UA: 5–406 μM	DA: 1 μM UA: 5 μM	DA: 97.67–102.16 UA: 97.81–102.89	[79]
LPE	MOF (ZIF-67)	sensor based on ZIF-67/PEDOT	A	DA in human blood	15–240 μM	0.04 μM	-	[80]
GPE	MXene (Ti_3_C_2_Cl_2_)	sensor based on IL-MXene	A	DA in human serum	100–2000 μM	702 nM	98.3–100	[81]
GCE	MXene (Ti_3_C_2_) MOF (UIO-66-NH_2_)	sensor based on Ti_3_C_2_/UIO-66-NH_2_	DPV	DA in human serum	1–250 fM	0.81 fM	101.2–103.5	[82]
AuE	MXene (Ti_3_C_2_T_x_)	sensor based on ZnO NPs/Ti_3_C_2_T_x_	CA	DA in human serum	0.1–1200 μM	0.076 μM	97.8–102.2	[83]
GCE	MXene (Ti_3_C_2_)	sensor based on Ti_3_C_2_/g-MWCNTs/ ZnO NSPs	DPV	DA in human serum	0.01–30 μM	3.2 nM	98.6–105.9	[84]
SPCE	MXene (Ti_3_C_2_)	sensor based on Ti_3_C_2_	DPV	DA in human urine Tyr in drug	0.5–600 μM	0.15 μM	DA: 97.1–104.0 Tyr: 96.7–102.5	[85]
GCE	TMD (MoS_2_)	sensor based on Ni-MoS_2_	DPV	DA in bovine serum	1 pM–1 mM	1 pM	97.0–105.0	[87]
FTO SPE	TMD (MoS_2_)	sensor based on MoS_2_	DPV	DA	Up to 300 μM	260 nM	-	[90]
GCE	MXene (Ti_3_C_2_T_x_)	sensor based on Tri-AgNPs/L-Cys/ MXene	DPV	5-HT in human serum	0.5–150 μM	0.08 μM	95.38–102.3	[92]
WGPE	TMD (WS_2_)	sensor based on WGP	DPV	DA, 5-HT in artificial CSF	DA: 1.21–13.37 μM 5-HT: 9.9–0.249 μM	DA: 1.24 μM 5-HT: 0.24 μM	95.2–104.1	[93]
GCE	MXene (Ti_3_C_2_)	sensor based on MXene/SWCNTs	CA	5-HT produced by living cells	0.004–103.2 μM	1.5 nM	-	[94]
GCE	MXene (Ti_3_C_2_)	sensor based on MXene/rGO	DPV	5-HT in human plasma	0.025–147 μM	10 nM	96.4–107	[95]

Abbreviations: A: amperometry; AuE: gold electrode; AuNPs: gold nanoparticles; CA: chronoamperometry; CC: carbon cloth; COF: covalent organic framework; CSF: cerebrospinal fluid; CV: cyclic voltammetry; DA: dopamine; DPV: differential pulse voltammetry; FTO: fluorine-doped tin oxide; GCE: glassy carbon electrode; GPE: graphite pencil electrode; 5-HT: 5-hydroxytryptamine (serotonine); IL: ionic liquid; LPE: lead pencil graphitic electrode; MOF: metal–organic framework; g-MWCNTs: graphitized multi-walled carbon nanotubes; NSPs: nanospheres; PGS: pyrolytic graphitic sheets; POM: polioxametalate; rGO: reduced graphene oxide; SPE: screen-printed electrode; SWCNTs: single-walled carbon nanotubes; TMD: transition metal dichalcogenide; Tri-AgNP: T triangular silver nanoplate; TS-COF: triazine-based covalent organic framework; Tyr: tyrosine; WGP: WS_2_/graphene heterostructure on polyimide.

**Table 3 molecules-29-00172-t003:** Analytical performances and format of electrochemical (bio)sensors for hormones determination.

Electrode	2D Nanomaterial	Format	Technique	Sample	Linearity	LOD	Recovery %	Reference Method	Ref.
Thread Conductive E	MXene	immunosensor immobilizing anti-cortisol on L-cys/AuNPs/ MXene	A	Cortisol in sweat	5–180 ng mL^−1^	0.54 ng mL^−1^	94.47–102	-	[99]
LBG/PDMS	MXene (Ti_3_C_2_T_x_)	immunosensorimmobilizing anti-cortisol on Ti_3_C_2_T_x_/LBG/ PDMS	EIS	Cortisol in sweat	0.01–100 nM	3.88 pM	93.68–99.1	-	[100]
ITOE	MXene (Ti_3_C_2_T_x_)	sensor based on GMA	DPV	EP in human urine	1–60 μM	3.5 nM	95.7–105.7	-	[102]
GCE	MOF (CoMn-ZIF)	sensor based on CoMnZIF-CNF	DPV	EP in human urine	5–1000 μM	1.667μM	97.51–102.53	-	[103]
DGE	MOF (ZIF-L)	aptasensor immobilizing insulin aptamer on Cu-BTC/ZIF-L	DPV	Insulin in human serum	0.1 pM–5 μM	0.027 pM	97.2–98.5	ELISA	[105]

Abbreviations: A: amperometry; AuNPs: gold nanoparticles; CNF: carbon nanofiber; Cu-BTC: copper(II) benzene-1,3,5-tricarboxylate; DGE: disposable gold electrode; DPV: differential pulse voltammetry; EIS: electrochemical impedance spectroscopy; ELISA: enzyme-linked immunosorbent assay; EP: epinephrine; GCE: glassy carbon electrode; GMA: reduced graphene oxide/MXene Ti_3_C_2_T_x_; ITOE: indium tin oxide electrode; L-cys: L-cysteine; LBG: laser graphene burned; MOF: metal–organic framework; PDMS: polydimethylsiloxane.

**Table 4 molecules-29-00172-t004:** Analytical performances and format of electrochemical (bio)sensors for pathogen determination.

Electrode	2D Nanomaterial	Format	Technique	Sample	Linearity	LOD	Recovery (%)	Reference Method	Ref.
SPCE	m-COF	immunosensor immobilizing IgY antibody on m-COF	SWV	*E. coli* in milk, beef, shrimps	10–10^8^ CFU mL^−1^	3 CFU mL^−1^	90–103	ELISA	[108]
IDEA	COP (CATN)	sensor based on CATN	EIS	*E. coli*	Up to 10 CFU mL^−1^	2 CFU mL^−1^	-	-	[109]
AuE	COF (Tph-TDC-COF)	aptasensor based on *E. coli* aptamer immobilized on Tph-TDC-COF	EIS/DPV	*E. coli* in bread, milk	10–10^8^ CFU mL^−1^	0.17 CFU mL^−1^ (EIS) 0.38 CFU mL^−1^ (DPV)	100.09–103.97 (bread) 99.61–100.71 (milk).	-	[111]
GCE	MXene	sensor based on MPBIP	EIS	*Salmonella* in drinking water	10^3^–10^7^ CFU mL^−1^	23 CFU mL^−1^	96–109.4	-	[112]
GCE	MXene (Ti_3_C_2_)	genosensor immobilizing ssDNA on PPY/MXene	DPV	*M.tb* in human sputum	100 fM- 25 nM	11.24 fM	90.52–100.8	PCR	[113]
MGCE	MXene (Ti_3_C_2_T_x_)	ECL immunosensor based on Fe_3_O_4_@Ab_1_-VV-R6G-Ti_3_C_2_T_x_@AuNRs-Ab_2_/ABEI	ECL	VV in seawater	1–10^8^ CFU mL^−1^	1 CFU mL^−1^	94.8–110.3	-	[115]
AuSPE	MXene	sandwich-type aptasensor involving PBA-Fc@Pt@MXenes	DPV	VP in shrimps	10–10^8^ CFU mL^−1^	5 CFU mL^−1^	95.0–104.3	-	[116]
AuE	MOF (MIL-53 (Al))	sandwich-type aptasensor using 2 aptamers immobilized on AuE and Au@Pt/MIL-53 (Al) nanocomposite modified with HRP and hemin/Gquadruplex DNAzyme as signal nanoprobe	DPV	SARS-CoV-2 NP	0.025–50 ng mL^−1^	8.33 pg mL^−1^	92–110	ELISA	[123]
SPCE	MOF (SiO_2_@UIO-66)	label-free immunosensor including SiO_2_@UiO-66 nanocomposite	EIS	SARS-CoV-2 SP in nasal fluid samples	100.0 fg∙mL^−1^–10.0 ng∙mL^−1^	100.0 fg mL^−1^	91.6–93.2	PCR	[124]
SPCE	MOF Ni_3_(BTC)_2_	label-free aptasensor using Ni_3_(BTC)_2_, SARS-CoV-2 aptamer, ePDA	EIS	SARS-CoV-2 virus in saliva, oropharyngeal swab	10–10^8^ PFU mL^−1^	3.3 PFU mL^−1^	98–104	PCR	[125]
IGE	MXene (Ti_3_C_2_T_x_)	genosensor using ssDNA/Ti_3_C_2_Tx	EIS	SARS-CoV-2 N gene	10^5^–10^9^ copies mL^−1^	10^3^ copies mL^−1^	-	-	[127]
GCE	Cu-MOF	genosensor including Cu-MOF/ErGO	DPV	HBV in human serum, urine	50.0 fM– 10.0 nM	5.2 fM	95.2–99.8	-	[128]
GCE	Cu-MOF	label-free immunosensor using Cu-MOF nanospheres	DPV	HBV in human serum	1–500 ng mL^−1^	730 pg mL^−1^	76.93–92.18	-	[129]

Abbreviations: Ab_1_: capture antibody; Ab_2_: detection antibody; ABEI: N-(4-aminobutyl)-N-ethylisoluminol; AuE: gold electrode; AuNRs: gold nanorods; AuSPE: gold screen-printed electrode; CATN: cationic network; COF: covalent organic framework; m-COF: magnetic COF; CFU: colony forming unit; COP: covalent organic polymer; DPV: differential pulse voltammetry; ELISA: enzyme-linked immunosorbent assay; ECL: electrochemiluminescence; EIS: electrochemical impedance spectroscopy; ePDA: electropolymerized poly(dopamine); ErGO: electrodreduced graphene oxide; Fc: ferrocene; GCE: glassy carbon electrode; HBV: *Hepatitis B virus*; HRP: horseradish peroxidase; IDEA: interdigitated electrode arrays; IGE: interdigitated gold electrode; L-cys: L-cysteine; LGB: laser graphene burned; MGCE: magnetic glassy carbon electrode; MOF: metal–organic framework; MPBIP: MXene/PPY-based bacterial imprinted polymer; *M.Tb*: *Mycobacterium tuberculosis*; PBA: phenylboronic acid; PCR: protein C reactive; PFU: plaque forming unit; PPY: poly(pyrrole); SPCE: screen-printed carbon electrode; ssDNA: single-stranded DNA; SPCE: screen printed carbon electrode; SWV: square wave voltammetry; Tph: 10,15,20-tetrakis(4-aminophenyl)-21H,23H-porphine); TDC: [2,2′-bithiophene]-2,5′-dicarbaldehyde; UiO: Universitetet Oslo; VP: *Vibrio parahaemolyticus*; VV: *Vibrio vulnificus*.

**Table 5 molecules-29-00172-t005:** Analytical performances and format of electrochemical (bio)sensors for cancer biomarkers determination.

Electrode	2D Nanomaterial	(Bio)Sensor Format	Technique	Sample	Linearity	LOD	Recovery%	Ref.
SPCE	MXene (Ti_3_C_2_)	Aptasensor including He@CCNT/Ti_3_C_2_	DPV	CEA in human serum	10–10^6^ pg mL^−1^	2.88 pg mL^−1^	95.29–105.19	[133]
FTOE	MXene (Ti_3_C_2_)	Immunosensor including Au-β-CD/MXene@ PANI	DPV	CEA in human serum	0.5–350 ng mL^−1^	42.9 pg mL^−1^	97.52–103.98	[134]
GCE	Mxene (Ti_3_C_2_T_x_)	Aptasensor involving Au-Pd-Pt/Ti_3_C_2_T_x_	DPV	CEA in human serum	1 fg mL^−1^ –1 ng mL^−1^	0.32 fg mL^−1^	98.8–104.1	[135]
GCE	COF-LZU1	Sandwich-type Immunosensor involving COF-LZU1, MrGOF, AgNPs	DPV	CA 125 in human serum	0.001–40 U mL^−1^	0.00023 U mL^−1^	91.54–105.21	[137]
GCE	COF BCN	Sandwich-type immunosensor involving MnO_2_@COF, AuPtNPs, BCN, PDA	DPV	PSA in human serum	0.00005–10 ng mL^−1^	16.7 fg mL^−1^	98.9–100.2	[138]
AuE	COF (p-COF)	Aptasensor based on EGFR aptamer immobilized on p-COF	DPV EIS	EGFR in human serum	0.05–100 pg mL^−1^	7.54 × 10^−3^ pg mL^−1^ (EIS) 5.64 × 10^−3^ pg mL^−1^ (DPV).	96.2–103.2	[139]
AuE	COF (p-COF)	Aptasensor based on EGFR aptamer immobilized on p-COF	EIS	MCF-7/-	5 × 10^2^–1 × 10^5^ cell·mL^−1^	61 cells·mL^−1^	-	[139]

Abbreviations: AgNPs: silver nanoparticles; AuPtNPs: gold platinum nanoparticles; AuE: gold electrode; β-CD: β-cyclodestrin; BCN: boron-doped carbon nitride; CEA: carcinoembryonic antigen; CNT: carbon nanotube; COF: covalent organic framework; DPV: differential pulse voltammetry; EGFR: epidermal growth factor receptor; He: herringbone-embedded; FTOE: fluorine-doped tin oxide electrode; GCE: glassy carbon electrode; MCF-7: Michigan Cancer Foundation-7; MrGOF: multi-layer reduced graphene oxide frame; p-COF: porphyrin-COF; PDA: polydopamine; PSA: prostate-specific antigen; SPCE: screen-printed carbon electrode; U: enzymatic unit.

**Table 6 molecules-29-00172-t006:** Analytical performances and format of electrochemical (bio)sensors for antibiotics determination.

Electrode	2D Nanomaterial	Format	Technique	Sample	Linearity	LOD	Recovery %	Reference Method	Ref.
AuE	Py-M-COF	Aptasensor including Py-M-COF	EIS	ENR in human serum	0.01–2000 pg mL^−1^	6.07 fg mL^−1^	101.0–112.4	-	[141]
AuE	Py-M-COF	Aptasensor including Py-M-COF	EIS	AMP in human serum	0.001–1000 pg mL^−1^	0.04 fg mL^−1^	99.5–103.0	-	[141]
GCE	Zr-amide-Por- based 2D COF	TC-MIECS	ECL	TC in milk	5–60 pM	2.3 pM	94.0–103.5	-	[142]
AuE	MOF (PCN-222)	Aptasensor including PCN-222/GO	EIS	CAP in milk, human serum, urine	0.01–50 ng mL^−1^	7.04 pg mL^−1^	94.6–107.2	-	[144]
GCE	COF	Electrochemical sensor based on Pd^2+^@P-CDPs/COFs	DPV	NF in eye drops	0.08–7.0 µM 7.0–100.0 µM	0.031 µM	97.8–101.7	HPLC	[145]
GCE	COF	Electrochemical sensor based on MIP/CuS/Au@COF	DPV	STZ in mutton, fodder, chicken liver, pig liver	0.001–100,000 nM	0.0043 nM	83.0–107.3	HPLC	[147]
CPE	MOF (ZIF-67)	Electrochemical sensor based on Fe_3_O_4_/ZIF-67/IL	DPV	Sulfamethoxazole in river, tap water, urine	0.01–520.0 μM	5.0 nM	97.1–103.3	-	[149]
GCE	COF	Electrochemical sensor based on GO@COF nanocomposite and MIP	DPV	SDZ/beef, pork, chicken, fodder	0.5–200 μM	0.16 μM	82.0–108.0	HPLC	[150]
AuE	MOF (PCN-222(Fe) NS)	Aptasensor including PCN-222 (Fe) NS	EIS	TOB/milk	1.1 × 10^−4^–10.7 nM	1.3 × 10^−4^ nM	97.5–104.8	-	[151]
AuE	COF (Tp-Bpy COF NS)	Aptasensor including Tp-Bpy COF NS	EIS	TOB/milk, river water	2.1 × 10^−4^–10.7 nM	6.57 fM	95.4–105.2% milk 96.1–104.4% river	-	[152]

Abbreviations: AMP: ampicillin; AuE: gold electrode; Bpy: 5,5′ -diamino-2,2′ -bipyridine; CAP: chloramphenicol; COF: covalent organic framework; CPE: carbon paste electrode; DPV: differential pulse voltammetry; ECL: electrochemiluminescence; EIS: electrochemical impedance spectroscopy; ENR: enrofloxacin; GCE: glassy carbon electrode; GO: graphene oxide; HPLC: high performance liquid chromatography; IL: ionic liquid; M: melamine; MOF: metal–organic framework; NF: norfloxacin; P-CDPs: β-cyclodextrin porous polymers; PCN-222 (Fe) NS: zirconium metal–organic framework nanosheet; Py: 1,3,6,8-tetrakis(4-formylphenyl)pyrene); SDZ: sulfadiazine; STZ: sulfathiazole; TC: tetracycline; TC-MIECS: TC-molecularly imprinted electrochemiluminescence sensor; TOB: tobramycin; Tp: 2,4,6-triformylphloroglucinol; Zr-amide-Por-based 2D COF: Zr-coordinated amide porphyrin-based 2D COF.

## Data Availability

Not applicable.

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
