# Peer review of "Not Only Graphene Two-Dimensional Nanomaterials: Recent Trends in Electrochemical (Bio)sensing Area for Biomedical and Healthcare Applications"

_molecules, 2023, doi:10.3390/molecules29010172_

Round 1
Reviewer 1 Report
Comments and Suggestions for Authors
- The abstract needs to reflect the technical advancements of the work rather than a description of the various sections of the review. It is suggested to include a brief scope of the topic, time period, the basis on which the papers reviewed are selected, and the major achievements highlighted in the literature reviewed.
-The manuscript needs to be checked for grammatical and spelling issues.
-“MXenes are classified as layered sheet-like nanomaterials and their thickness starting from 1 mm". Do the authors mean 1mm or 1 nm?
-Quantum dots that are derived from 2D nanomaterials, such as graphene quantum dots, are not considered 2D, due to their small size.
-More references need to be cited in the sections where synthesis approaches are discussed. The following references describe the synthesis of 2D nanomaterials. I recommend citing them and add more citations referring to the synthesis of 2D nanomaterials.
https://doi.org/10.1016/j.molliq.2020.113087
https://doi.org/10.1021/acsaem.8b00838
https://doi.org/10.1016/j.colsurfa.2021.126793
-Authors have mentioned “In addition, nanomaterials and in particular 2D nanomaterials can be considered as new bioreceptors, acting both as bioreceptors”. I don’t think 2D nanomaterials are bioreceptors.
-I suggest to group papers by concept, chronology of ideas and critically analyze key contributions and findings.
-I recommend to group examples based on the design criteria, instead of reviewing several unrelated examples. Taking glucose as an example, I cannot establish any connection between references 46, 47 and 48. What have been the challenges? Which challenges are improved by modification?
-Avoid using adverbs such as unfortunately and interesting
-Authors must focus on the synthesis approaches of 2D MOFs rather than MOFs, in general, as MOFs are considered as 3D porous structures.
-In each example, authors must clarify what is the role of 2D nanomaterial regarding the performance of the biosensor.
-In reference 45, there is no 2D nanomaterial involved.
Comments on the Quality of English Language-The manuscript needs to be checked for grammatical and spelling issues.
Author Response
Our response to reviewer 1 is in the attached file

Reviewer 2 Report
Comments and Suggestions for Authors
The manuscript “Not Only Graphene Two-Dimensional Nanomaterials: Recent Trends in Electrochemical (Bio)sensing Area for Biomedical and Healthcare Applications” is a timely and interesting review. The authors focus on the application of various two-dimensional nanomaterials in electrochemical biosensing and sensing systems for the detection of crucial analytes in clinical and biomedical contexts. The advantages, limitations, and future prospects of these materials were also discussed. In general, this manuscript merits publication in Molecules. However, there are still some areas that require improvement before final acceptance.
1. The manuscript has some minor formatting issues, like the citation on line 391 and the table mentioned on line 834. Hence, the author should carefully review the manuscript to rectify these simple errors.
2. The exposition in both the introduction and conclusion of the manuscript lacks clarity, and an excessive number of paragraphs contributes to the overall lack of lucidity in the manuscript, thereby diminishing the article's readability. At the same time, the manuscript is lengthy, and it is suggested that careful reduction should be considered.
3. In the third section, where the properties and manufacturing methods of two-dimensional materials are discussed, the synthesis part is excessively detailed and fails to sufficiently emphasize their advantages in the field of electrochemical sensing.
4. The examples in the manuscript lack proper prioritization and organization. For instance, in the section discussing electrochemical methods for glucose, there are over a dozen examples presented without sufficient summarization and analysis.
Comments on the Quality of English Language
The English of the manuscript is readible, but the sentences needs to be further refined.
Author Response
Our response to reviewer 2 is in the attached file

Reviewer 3 Report
Comments and Suggestions for Authors
This review focuses on the application of different 2D nanomaterials beyond graphene in electrochemical biosensing and sensing systems to detect significant analytes of clinical and biomedical interest. I think this review may be published after major revision. Some comments are as follows:
(1) The authors need to reorganize the introduction, which is not only very important to highlight the significance of this review but also is beneficial for readers to understand the development status of this field.
(2) Some reviews about biosensors have been reported. The authors should cite and compare with these reviews to highlight the meaningful of this work, including Molecules 28 (2023) 4617, Trac-Trends in Analytical Chemistry 162 (2023) 117027, Dalton Transactions 50 (2021) 14091 and so on.
(3) In the part of COFs, some COFs-based electrochemical biosensors should be cited and described, such as Sensors and Actuators B-Chemical 313 (2020) 128033, Biosensors & Bioelectronics 126 (2019) 734, Journal of Electroanalytical Chemistry 881 (2021) 114931, Microchimica Acta 190 (2023) 421, Analytica Chimica Acta 1223 (2022) 340204, Sensors 22 (2022) 4758, Sensors and Actuators B-Chemical 300 (2019) 126993.
(4) MOFs and their hybrid materials as 2D functional materials have been widely used to fabricate electrochemical sensors, so the authors should cite and compare these reports, such as ACS Applied Materials & Interfaces 15 (2023) 16991, Talanta 268 (2023) 125344, Microporous and Mesoporous Materials 323 (2021) 111200, Journal of Materials Chemistry C 8 (2020) 15823, Inorganic Chemistry Communications 145 (2022) 109970, Journal of Electroanalytical Chemistry 916 (2022) 116382, Journal of the Electrochemical Society 169 (2022) 046502, Sensors and Actuators B-Chemical 368 (2022) 132129.
(5) Pesticide should be mentioned in the part of 3. Applications of 2D nanomaterials to electrochemical (bio)sensors for healthcare and biomedical field, such as Journal of the Electrochemical Society 168 (2021) 037504, Chemical Engineering Journal 433 (2022) 133639, ACS Sensors 7 (2022) 3551, Journal of Electroanalytical Chemistry 909 (2022) 116115, Journal of the Electrochemical Society 165 (2018) B848.
Comments on the Quality of English LanguagePlease check the article carefully to avoid errors.
Author Response
Our response to reviewer 3 is in the attached file

Round 2
Reviewer 1 Report
Comments and Suggestions for Authors
In the electrochemical biosensors, 2D nanomaterials mostly act as the transducer or carriers of signal elements. I do not consider them as synthetic receptors. Synthetic receptors are designed and synthesized to mimic the binding properties of natural receptors like antibodies, but with advantages of robustness and stability. Some examples are MIPs, aptamer, and peptides. If authors insist on the fact that 2D nanomaterials are receptors, they need to cite some references to prove their idea.
In the original article of reference 50, it is mentioned that “ZIF-67 has a well-defined dodecahedron with a uniform size of about 200 nm”. Please note that ZIF-67 can be synthesized in different morphologies by changing the solvent (10.1021/acsami.0c11269).
Author Response
Our response to reviewer 1 are present in the attached file

Reviewer 2 Report
Comments and Suggestions for Authors
The authors have addressed my concerns, and it is recommended for publication now.
Comments on the Quality of English LanguageThe quality of English in the revised version is acceptable.
Author Response
We would like to thank the reviewer for his/her support
Reviewer 3 Report
Comments and Suggestions for Authors
I think the revised manuscript can be published in this journal.
Author Response

(The authors gave the same response as above.)
